# G protein-coupled receptor kinase-2 (GRK-2) controls exploration through neuropeptide signaling in *Caenorhabditis elegans*

Kristen Davis[1,2,3], Christo Mitchell[4☯], Olivia Weissenfels[4☯], Jihong Bai[5], David M. Raizen[1,6], Michael Ailion☉[4]*, Irini Topalidou☉[4,5]*

1 Department of Neurology, Perelman School of Medicine, University of Pennsylvania, Philadelphia, Pennsylvania, United States of America, 2 Center for Excellence in Environmental Toxicology (CEET), Perelman School of Medicine, University of Pennsylvania, Philadelphia, Pennsylvania, United States of America, 3 Department of Neuroscience, Vickie and Jack Farber Institute of Neuroscience, Thomas Jefferson University, Philadelphia, Pennsylvania, United States of America, 4 Department of Biochemistry, University of Washington, Seattle, Washington, United States of America, 5 Fred Hutchinson Cancer Center, Seattle, Washington, United States of America, 6 Chronobiology and Sleep Institute, Perelman School of Medicine, University of Pennsylvania, Philadelphia, Pennsylvania, United States of America

☯ These authors contributed equally to this work.
* mailion@uw.edu (MA); etopalid@fredhutch.org (IT)

**Data Availability Statement:** All relevant data are within the manuscript and its Supporting information files.

## Abstract

Animals alter their behavior in manners that depend on environmental conditions as well as their developmental and metabolic states. For example, *C. elegans* is quiescent during larval molts or during conditions of satiety. By contrast, worms enter an exploration state when removed from food. Sensory perception influences movement quiescence (defined as a lack of body movement), as well as the expression of additional locomotor states in *C. elegans* that are associated with increased or reduced locomotion activity, such as roaming (exploration behavior) and dwelling (local search). Here we find that movement quiescence is enhanced, and exploration behavior is reduced in G protein-coupled receptor kinase *grk-2* mutant animals. *grk-2* was previously shown to act in chemosensation, locomotion, and egg-laying behaviors. Using neuron-specific rescuing experiments, we show that GRK-2 acts in multiple ciliated chemosensory neurons to control exploration behavior. *grk-2* acts in opposite ways from the cGMP-dependent protein kinase gene *egl-4* to control movement quiescence and exploration behavior. Analysis of mutants with defects in ciliated sensory neurons indicates that *grk-2* and the cilium-structure mutants act in the same pathway to control exploration behavior. We find that GRK-2 controls exploration behavior in an opposite manner from the neuropeptide receptor NPR-1 and the neuropeptides FLP-1 and FLP-18. Finally, we show that secretion of the FLP-1 neuropeptide is negatively regulated by GRK-2 and that overexpression of FLP-1 reduces exploration behavior. These results define neurons and molecular pathways that modulate movement quiescence and exploration behavior.

**Funding:** This work was supported by NIH Grants R01s NS109476 to MA and JB, GM121481 to MA, NS107969 and NS122779 to DMR. DK was supported by a NIH T32Act ES019851 Training Grant. The funders had no role in study design, data collection and analysis, decision to publish, or preparation of the manuscript.

**Competing interests:** The authors have declared that no competing interests exist.

## Author summary

Many modulatory neurotransmitters affect behavior by binding to G protein-coupled receptors (GPCRs) and initiating signals that modify neuronal activity. GPCRs are regulated by G protein-coupled receptor kinases (GRKs). GRKs phosphorylate and promote the inactivation of GPCRs. Here we identify GRK-2 as a regulator of distinct locomotor states in *C. elegans*. We find that GRK-2 acts in olfactory sensory neurons to promote exploration and suppress movement quiescence. Additionally, we show that GRK-2 acts in opposition to a neuropeptide signaling pathway that acts in interneurons. Thus, this study demonstrates critical roles for GRK-2 in regulating neuromodulatory signaling and locomotor behavior.

## Introduction

Animals modify their behaviors in response to changes in their environment or metabolic state. Sensory and neuromodulatory signals play a major role in adapting behavior to environmental or metabolic changes [1–6]. For example, animals experience periods of quiescence and arousal that depend on sensory and neuromodulatory cues [3,5,7–9]. Arousal is a state of increased responsiveness that can be associated with fear or starvation, whereas quiescence is characterized by decreased responsiveness and is associated with sleep and satiety. Even though sensory and neuromodulatory signals are recognized as major modulators of behavior, the precise mechanisms that mediate behavioral strategies under different inputs remain largely unknown. Here we describe how a locomotor behavior in the nematode *Caenorhabditis elegans* is altered through the action of the G protein-coupled receptor kinase-2 (GRK-2) in sensory neurons and neuromodulator signaling in interneurons.

Several examples of movement quiescence in response to changes in the external environment or the internal metabolic state of the animal have been reported in *C. elegans*. *C. elegans* enters an exploration state when it is removed from food [10,11] and becomes quiescent during molting or under conditions of prolonged starvation, satiety, or cellular stress [3,12–17]. Sensory perception affects quiescence, as well as additional locomotor states in *C. elegans* that are characterized by increased or reduced locomotion activity, such as roaming/exploration and dwelling [2]. The animal covers a large area during roaming (exploring), while it restricts its activity to a small region during the dwelling state. The mechanism by which sensory perception modulates this locomotor strategy is unknown.

Quiescence, roaming, and dwelling are controlled by the function of the cGMP-dependent protein kinase (PKG) gene *egl-4* in sensory neurons. *egl-4* loss-of-function mutant animals have chemosensory defects [18] and attenuated sensory adaptation [19]. EGL-4 is a regulator of lethargus quiescence, a sleep-like behavior that takes place during molts [3], and of a satiety-induced sleep-like behavior [12]. EGL-4 also defines the time the animals spend roaming and dwelling since *egl-4* loss of function mutant animals show increased exploration [2,20]. Here, we describe a role for GRK-2 in the sensory neurons of *C. elegans* to regulate movement quiescence and exploration in an opposite manner to EGL-4.

GRKs play key roles in attenuating the strength of neuromodulatory pathways by phosphorylating G protein-coupled receptors (GPCRs) and promoting the inactivation of GPCR signaling [21]. Vertebrates have seven GRKs that are classified into three subfamilies: the visual GRKs (GRK1/7), the GRK2/3 subfamily, and the GRK4/5/6 subfamily [22]. *C. elegans*, which lacks a visual system, has GRK-1 and GRK-2, which are orthologs of the mammalian GRK4/5/6 and GRK2/3 families, respectively [23,24]. *grk-2* is expressed broadly in the nervous system

and has a broad impact on animal physiology including chemosensation [23], gustatory plasticity [25], egg-laying [26], crawling [27], swimming [27,28], and oxidative and heat stress responses [29,30]. Elimination of *grk-2* function renders animals unable to respond to a wide range of chemical stimuli, suggesting that GRK-2 plays a positive role in chemosensation [23]. Similarly, *grk-2* mutant animals show reduced crawling rates and are unable to swim, again suggesting that GRK-2 plays a positive role in regulating crawling and swimming [27,28]. GRK-2 was proposed to promote crawling and swimming by negatively regulating the D2-like dopamine receptor DOP-3 [27,28]. GRK-2 regulates chemosensation by acting in sensory neurons [23], but it controls crawling and swimming by acting in premotor interneurons [27,28], suggesting that GRK-2 acts in different neurons to modulate distinct behavioral outcomes.

Besides sensory perception, neuromodulation has also been shown to play a critical role in defining arousal and quiescence as well as roaming and dwelling in *C. elegans*. For example, mutants lacking the neuropeptide receptor NPR-1 have been used as a model for studying arousal since both lethargus and movement quiescence are reduced in *npr-1* mutants [8,31–33]. Specifically, increased RMG interneuron activity in *npr-1* mutant animals during lethargus induces secretion of the neuropeptide pigment dispersing factor PDF-1, resulting in increased animal motility and reduced quiescence [31]. PDF-1 also positively regulates roaming, whereas serotonin plays an opposite role by increasing dwelling [4]. Here we find that GRK-2 controls exploration in an opposite manner to NPR-1 and the neuropeptides FLP-1 and FLP-18. We also show that GRK-2 acts in different pathways than PDF and serotonergic signaling. Finally, we demonstrate that secretion of FLP-1, a neuropeptide that regulates locomotion [34–37], depends on GRK-2 and that overexpression of *flp-1* inhibits exploration behavior.

## Results

### *grk-2* mutants have exploration and dispersal defects

Our previous work suggested that GRK-2 positively regulates crawling and swimming by negatively regulating the D2-like dopamine receptor DOP-3 [27,28]. *grk-2(gk268)* null mutant animals exhibit slow crawling and impaired swimming, and mutations in *dop-3* suppress both these phenotypes. *grk-2* mutant animals also have a decreased exploration behavior: they restrict their movements to a limited region of the bacterial lawn, whereas wild-type animals explore the entire lawn [27]. This exploration behavior defect could be due to decreased roaming, increased dwelling, or increased quiescence. We used simple radial locomotion and exploration assays ([4,38]; see also Methods) to quantify this phenotype by observing the radial distance that individual animals traveled over 1 h (Fig 1A) or the tracks that individual animals left on a lawn of *E. coli* over an 18–20 hr period (Fig 1D). We found that *grk-2(gk268)* mutant animals disperse and explore less compared to wild-type (Fig 1B and 1E). This phenotype is independent of sex since *grk-2* males also explore less than wild type males (Fig 1F). To examine whether the *grk-2* dispersal defect depends on the presence of food we performed the dispersal assay on plates without *E. coli*. We found that *grk-2 (gk268)* mutant animals disperse less compared to wild type even in the absence of the bacteria lawn (Fig 1C), suggesting that the dispersal defect of *grk-2* mutant is independent of the presence or absence of food.

Although the *dop-3(vs106)* mutation suppresses the slow locomotion and swimming defects of *grk-2(gk268)* mutant animals [27,28], it did not significantly alter the dispersal or exploration behavior of *grk-2(gk268)* (Fig 1B, 1C and 1E), suggesting that GRK-2 affects the exploration behavior independently of DOP-3, and that the mechanism by which GRK-2 influences exploration behavior is different than its role in regulating crawling and swimming.

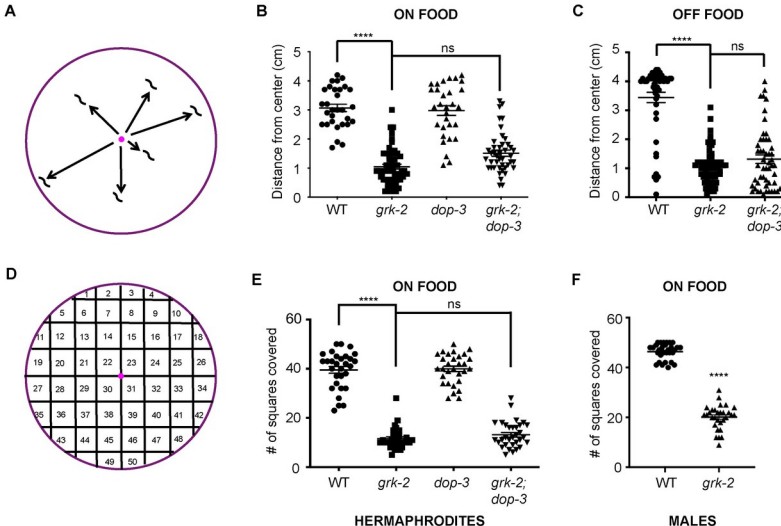

**Fig 1. *grk-2* mutant animals have dispersal and exploration defects.** (A) Schematic representation of the assay used to measure dispersal (see Methods). (B) *grk-2(gk268)* mutant animals have a dispersal defect in the presence of food and *dop-3(vs106)* does not rescue this phenotype. Shown is the distance (cm) animals traveled from the center of the plate in the presence of food in 1 h. (****, P<0.0001. ns, P>0.05. One way ANOVA followed by a Bonferroni test. Error bars = SEM; n = 30–53). (C) *grk-2(gk268)* mutant animals have a dispersal defect in the absence of food and *dop-3(vs106)* does not rescue this phenotype. Shown is the distance (cm) animals traveled from the center of the plate without food in 20 min. (****, P<0.0001. ns, P>0.05. One way ANOVA followed by a Bonferroni test. Error bars = SEM; n = 47–60). (D) Schematic representation of the assay used to measure exploration (see Methods). (E) *grk-2(gk268)* mutant hermaphrodites have an exploration defect and *dop-3(vs106)* does not rescue this defect. Shown is the number of squares (out of 50) single animals covered in approximately 20 h. (****, P<0.0001. ns, P>0.05. One way ANOVA followed by a Bonferroni test. Error bars = SEM; n = 30–32). (F) *grk-2(gk268)* mutant males have an exploration defect. (****, P<0.0001. Student's t-test. Error bars = SEM; n = 30).

## GRK-2 acts in a different pathway than serotonergic or PDF signaling to affect exploration

Worm exploration behavior is regulated by serotonergic and PDF signaling: serotonin inhibits exploration through the MOD-1 serotonin-gated chloride channel, while PDF-1 promotes exploration through the Gαs-coupled PDF receptor PDFR-1 [4]. To examine whether GRK-2 acts in the PDF-1 pathway to affect the exploration behavior we compared animals that were mutant for both *grk-2* and *pdfr-1* to single mutants. We found that *pdfr-1(ok3425)* mutant animals display decreased exploration, as previously reported [4] and that the phenotype of the double mutant *grk-2(gk268); pdfr-1(ok3425)* is stronger than the phenotype of the single mutants (S1A Fig), suggesting that GRK-2 and PDFR-1 act in different pathways to control exploration. Similarly, to examine whether GRK-2 acts in the serotonin pathway to control exploration, we built double mutants between *grk-2* and *tph-1* mutants (*tph-1* encodes the rate-limiting enzyme for serotonin synthesis). We verified that *tph-1(mg280)* mutant animals display increased exploration [4] and further showed that *grk-2(gk268); tph-1(mg280)* double mutant animals have an intermediate phenotype between that of either *grk-2* or *tph-1* single mutants (S1B Fig), suggesting that *grk-2* and *tph-1* act in separate pathways to control exploration. Our results imply that GRK-2 acts in a different pathway than serotonergic or PDF signaling to affect the exploration behavior.

## GRK-2 functions in the ciliated neurons to affect the exploration behavior

GRK-2 is expressed broadly in the nervous system [23] and acts in premotor interneurons to control crawling and swimming [27,28]. To determine where GRK-2 acts to affect the

exploration behavior, we expressed the *grk-2* cDNA under the control of neuron-specific promoters. Expression of *grk-2* cDNA under the pan-neuronal promoter (*rab-3p*) fully rescued the *grk-2(gk268)* mutant exploration defect but expression in ventral cord acetylcholine motor neurons alone (using the *acr-2* promoter) or in command interneurons alone (using the *nmr-1* promoter) did not rescue this defect (Fig 2A). Expression of *grk-2* cDNA in the same set of command interneurons rescued the crawling defects of *grk-2* mutant animals (S2 Fig), supporting the notion that the role that GRK-2 plays to affect the exploration behavior is different from its role in crawling.

 *grk-2* mutants fail to respond to a wide range of chemical stimuli, suggesting that GRK-2 is a positive regulator of chemosensation [23]. To test the hypothesis that the exploration defect in *grk-2(gk268)* mutant animals is explained by its role in chemosensation, we expressed the *grk-2* cDNA under the ciliated sensory neuron promoters *xbx-1p* and *osm-6p*. Expression of *grk-2* cDNA in ciliated sensory neurons fully rescued the exploration defect of *grk-2* mutant animals both in the presence and absence of food (Fig 2B and 2C). Our result that transgenic expression of *grk-2* under *rab-3*, *xbx-1*, or *osm-6* heterologous promoters fully rescued the *grk-2* exploration defect (Fig 2A–2C) suggests that the precise level of GRK-2 is not critical to its function.

 To further narrow down the site of action of GRK-2 in exploration behavior we expressed the *grk-2* cDNA under the control of the *odr-3* promoter (*odr-3p*), which promotes expression in olfactory and nociceptive neurons: AWA, AWB, AWC, ASH, and ADF [39]. Expression of *grk-2* under the *odr-3* promoter partially rescued the exploration defect of *grk-2* mutant animals, suggesting that *grk-2* acts in multiple ciliated neurons to affect exploration behavior (Fig 2D). An operon GFP was included in the expression construct downstream of the 3'UTR allowing confirmation of expression (see Methods). To confirm that *grk-2* is expressed in *odr-3*- expressing neurons, we expressed the cDNA encoding GRK-2 protein fused to tagRFP (*grk-2*::tagRFP) under the *grk-2* promoter as well as mNeon::NLS driven by the *odr-3* promoter (*odr-3p*::mNeon::NLS). We confirmed that *grk-2* is indeed expressed in *odr-3*-expressing neurons (shown with arrows in Fig 2G).

 To dissect further the subset of chemosensory neurons where GRK-2 acts to control exploration behavior we expressed the *grk-2* cDNA under the control of the *sra-6* promoter, which is primarily expressed in the ASH and ASI chemosensory neurons (*sra-6p*; [40]). Expression of *grk-2* under the *sra-6* promoter partially rescued the *grk-2* exploration defect, suggesting that a part of GRK-2's role in exploration behavior is due to its action in the ASH or ASI neurons (Fig 2E). Similarly, *grk-2* expression in the AWA, AWB, or AWC neurons (*odr-10p*: expression in AWA [41]; *str-1p*: expression in AWB [42]; *str-2p*: expression in AWC-ON [43]) only slightly rescued the *grk-2* exploration defect (Fig 2F). Even *grk-2* expression under the combined control of the *odr-1*, *str-1*, and *str-2* promoters only partially rescued the *grk-2* exploration defect (Fig 2E), again suggesting that *grk-2* acts in more than a single type of ciliated neuron to control exploration behavior. To address the possibility that expression of *grk-2* in any ciliated sensory neuron might rescue the exploration defect of *grk-2* mutant animals, we expressed *grk-2* under the *gcy-8* promoter, which is solely expressed in the AFD thermosensory neurons (*gcy-8p*; [44]). Expression of *grk-2* under the *gcy-8* promoter failed to rescue the *grk-2* exploration defect (Fig 2H). Similarly, expression of *grk-2* under the dopaminergic neuron promoter *dat-1* [45] failed to rescue the *grk-2* exploration defect (*dat-1p*; Fig 2I).

 To examine whether GRK-2 plays a developmental role to affect exploration we placed *grk-2* cDNA under the control of a heat shock-inducible promoter and introduced it in *grk-2* mutant animals. Heat shock in adult animals partially but significantly restored the exploration behavior of *grk-2* (Fig 2J). These results together suggest that GRK-2 has a functional role in mature ciliated sensory neurons that affects the exploration behavior.

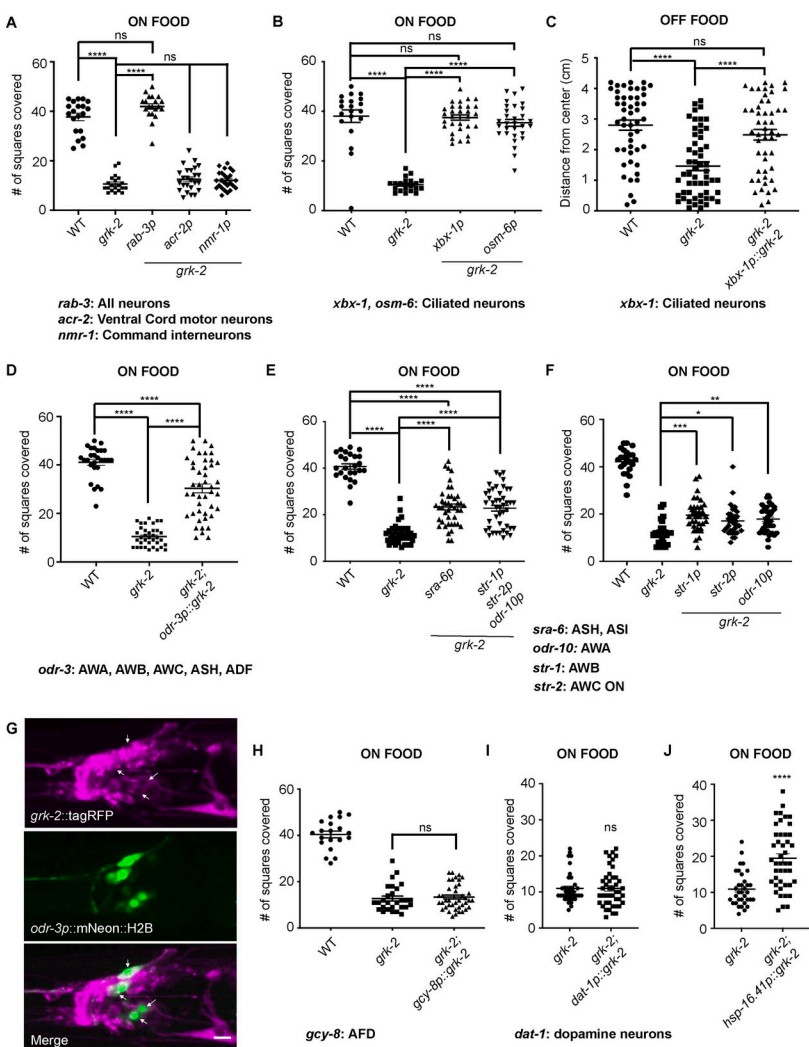

**Fig 2. *grk-2* acts in the ciliated neurons to control exploration behavior.** (A) Expression of *grk-2* cDNA under a pan-neuronal promoter rescues the exploration defect of *grk-2* mutant animals in the presence of food. The *grk-2* cDNA was expressed in *grk-2(gk268)* mutant animals under a pan-neuronal promoter (*rab-3p*, transgene *yakEx44*), ventral cord acetylcholine motor neuron promoter (*acr-2p*, transgene *yakEx47*), and interneuron promoter (*nmr-1p*, transgene *yakSi32*). Shown is the number of squares single animals covered in ~20 h. (****, P<0.0001. ns, P>0.05. One way ANOVA followed by a Bonferroni test. Error bars = SEM; n = 20–25). (B) Expression of *grk-2* cDNA under either of two ciliated sensory neuron promoters (*xbx-1p*, transgene *yakEx71* and *osm-6p*, transgene *yakEx53*) rescues the exploration defect of *grk-2(gk268)* mutant animals in the presence of food. (****, P<0.001. ns, P>0.05. One way ANOVA followed by a Bonferroni test. Error bars = SEM; n = 20–30). (C) Expression of *grk-2* cDNA under a ciliated sensory neuron promoter (*xbx-1p::grk-2*, transgene *yakEx71*) rescues the dispersal defect of *grk-2(gk268)* in the absence of food. (****, P<0.0001. ns, P>0.05. One way ANOVA followed by a Bonferroni test. Error bars = SEM; n = 49–55). Please note the high variability in the off-food response in comparison to the on-food response (Fig 2B). This could be because worms are more mobile in the absence of food and constantly searching for a food source, which would increase the variability of the distances they cover. (D) Expression of *grk-2* cDNA under the *odr-3* promoter (expression in AWA, AWB, AWC, ASH, and ADF neurons; *odr-3p::grk-2*, transgene *yakEx189*) partially rescues the exploration defect of *grk-2(gk268)*. (****, P<0.0001. One way ANOVA followed by a Bonferroni test. Error bars = SEM; n = 27–44). (E) Expression of *grk-2* cDNA under the *sra-6* promoter (expression in ASH and ASI neurons; *sra-6p::grk-2*, transgene *yakEx191*) or under the combined control of the *odr-10* (AWA neurons), *str-1* (AWB neurons) and *str-2* (AWC ON neuron) promoters (*str-1p*, *str-2p*, *odr-10p::grk-2*, transgene *yakEx194*) partially rescues the exploration defect of *grk-2(gk268)*. (****, P<0.0001. One way ANOVA followed by a Bonferroni test. Error bars = SEM; n = 25–44). (F) Expression of *grk-2* cDNA under the *odr-10p* (AWA), *str-1p* (AWB), or *str-2p* (AWC ON) promoters (transgenes *pekEx267*, *pekEx265* and *pekEx266*, respectively) partially rescues the *grk-2* exploration defect. (***, P<0.001. **, P<0.01. *, P<0.05. One way ANOVA followed by a Bonferroni test. Error bars = SEM; n = 20–33). Twenty of the data points shown for WT and *grk-2* mutant are the same as in Fig 3B (these experiments were run in

parallel). (G) *grk-2* is expressed in many neurons in the head of the worm including *odr-3*- expressing neurons. Representative images of the z-stack projections of the head of an animal co-expressing tagRFP fused to GRK-2 ORF driven by the *grk-2* promoter (*grk-2*::tagRFP, transgene *yakIs19*) and NLS-tagged mNeon driven by the *odr-3* promoter (*odr-3p*::mNeon::NLS; transgene *yakEx204*). Arrows indicate *grk-2*::tagRFP-expressing cells that also express *odr-3p*::mNeon::NLS. Scale bar: 10 μm. (H) Expression of *grk-2* cDNA under the *gcy-8* promoter (expression in AFD neurons; *gcy-8p*::*grk-2*, transgene *yakEx202*) fails to rescue the exploration defect of *grk-2(gk268)*. (ns, P>0.05. One way ANOVA followed by a Bonferroni test. Error bars = SEM; n = 20–40). (I) Expression of *grk-2* cDNA under the *dat-1* promoter (expression in dopaminergic neurons; *dat-1p*::*grk-2*, transgene *yakEx259*) fails to rescue the exploration defect of *grk-2(gk268)*. (ns, P>0.05. Student's t-test. Error bars = SEM; n = 38–44). (J) Expression of *grk-2* cDNA under the heat shock promoter *hsp-16.41* (*hsp-16.41p*::*grk-2*, transgene *yakEx253*) in adult *grk-2(gk268)* partially but significantly rescues their exploration defect. (****, P<0.0001. Student's t-test. Error bars = SEM; n = 34–44).

## Mutants in cilia-related genes have exploration defects similar to *grk-2*

Mutants with defects in sensory perception exhibit exploration defects on food like the ones described here for *grk-2* [2]. Specifically, mutants with cilium-structure defects (such as *che-2*) restrict their movements to a limited region of the bacterial lawn, while sensory signal transduction mutants like *tax-2* and *tax-4* have a weaker defect than cilium-structure mutants [2]. *grk-2* mutant animals are not defective in their ability to take up a fluorescent dye like the cilium-structure mutants (Dye-filling assay; [23]), an assay indicative of the structural integrity of the sensory cilia. To confirm that cilium structure mutants have exploration defects similar to *grk-2*, we performed the exploration assay on *che-2(e1033)* and *daf-10(e1387)* mutant animals in the presence or absence of food. We found that *che-2(e1033)* and *daf-10(e1387)* mutant animals are exploration defective identical to *grk-2(gk268)* (Fig 3A–3C). Additionally, the *che-2(e1033)* and *daf-10(e1387)* mutations do not enhance the phenotype of *grk-2* mutant (Fig 3A–3C). The lack of an obvious enhancement is not due to a floor effect since *grk-2; pdfr-1* double mutant animals have stronger exploration phenotype than the single mutants (S1A Fig). These findings suggest that *che-2*, *daf-10*, and *grk-2* act in the same pathway to control exploration (Fig 3A–3C).

To further investigate which kinds of sensory perception affect exploration, we used the exploration assay to test mutants with different sensory defects. Most chemosensory and osmosensory- defective mutants that possess defective cilia based on their Dye filling phenotype [46] had defects in exploration (Fig 3D). Specifically, mutants with strong cilium-structure defects (shown in blue in Fig 3D) had strong exploration defects and most mutants with weak cilium defects (shown in green in Fig 3D) had weaker exploration defects. Other chemosensory, osmosensory, and thermosensory defective mutants (*odr-1*, *odr-3*, *odr-8*, *odr-10*, *osm-7*, *osm-10*, *tax-2*, *tax-4*, *ttx-1(p767)*), also had exploration defects but not as severe as *grk-2*. Interestingly, chemosensory and osmosensory mutants *che-1*, *che-6*, *che-7*, *odr-2*, *odr-4*, *odr-5*, *odr-7*, *osm-9*, and *osm-11*, as well as thermosensory *ttx-1(oy26* and *oy29)* mutants were not significantly defective. Surprisingly, the *ocr-2* canonical null mutant *ak47* (VM396 strain) showed an exploration defect that was not observed with the *ocr-2(ok1711)* allele (VC1233 strain). Given that *ok1711* contains a deletion of at least three exons it is presumably a null allele, suggesting that either the VC1233 *ocr-2(ok1711)* strain has a suppressor mutation that rescues the exploration defect or that the VM396 *ocr-2(ak47)* strain has an exploration defect due to a background mutation and not because of the *ocr-2(ak47)* deletion. To distinguish between these possibilities, we outcrossed strains VC1233 and VM396 and tested the exploration behavior of outcrossed strains BJH2762 *ocr-2(ok1711)* and BJH2763 *ocr-2(ak47)*. We found that strain BJH2763 *ocr-2(ak47)* did not have an exploration defect, suggesting that the *ocr-2 (ak47)* mutation does not cause exploration defects and that strain VM396 has a background mutation that makes the animals exploration defective (S3A Fig). Notably, oxygen-sensing mutant *gcy-35* and mechanosensory-defective mutants *mec-3*, *mec-4*, and *glr-1* did not show

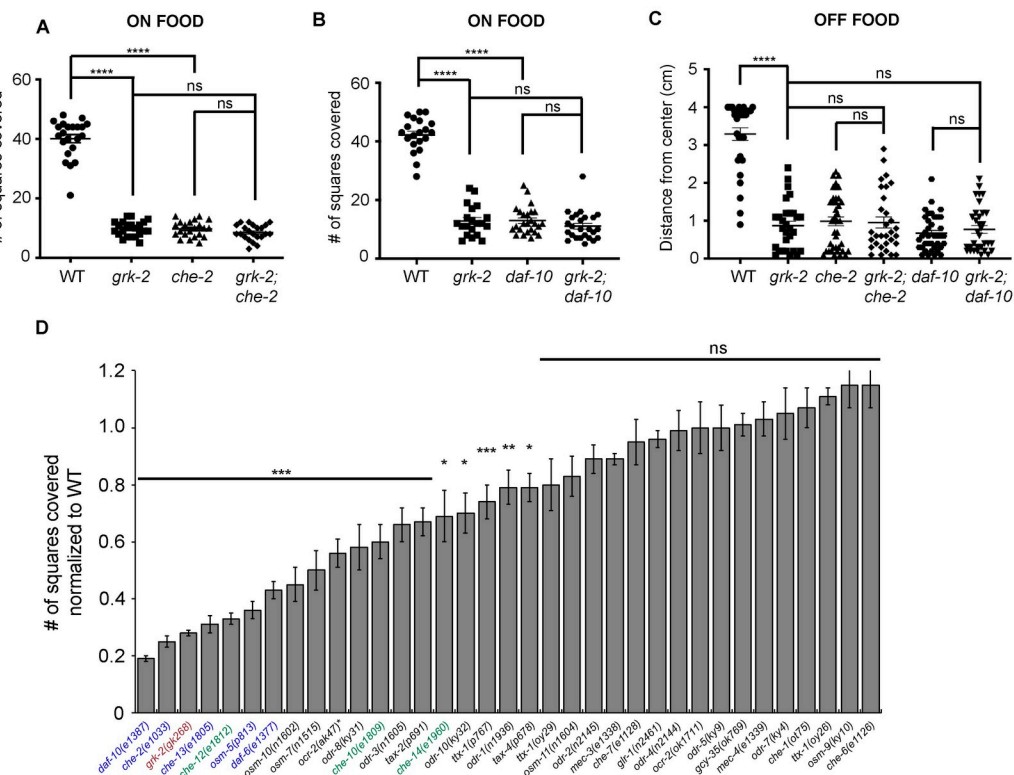

**Fig 3. Mutants in cilia-related genes have exploration defects.** (A) The *che-2(e1033)* mutant has an exploration defect like *grk-2(gk268)* that is not enhanced by *grk-2*. Shown is the number of squares single animals covered in ~20 h. (\*\*\*\*, P<0.0001. ns, P>0.05. One way ANOVA followed by a Bonferroni test. Error bars = SEM; n = 22). (B) The *daf-10(e1387)* mutant has an exploration defect like *grk-2(gk268)* that is not enhanced by *grk-2*. Shown is the number of squares single animals covered in ~20 h. (\*\*\*\*, P<0.0001. ns, P>0.05. One way ANOVA followed by a Bonferroni test. Error bars = SEM; n = 20–25). The data points shown for WT and *grk-2* mutant are the same as in Fig 2F (these experiments were run in parallel). (C) The *che-2(e1033)* and *daf-10(e1387)* mutants have a dispersal defect independently of food that is not enhanced by *grk-2(gk268)*. Shown is the distance (cm) animals traveled from the center of the plate without food in 20 min. (\*\*\*\*, P<0.0001. One way ANOVA followed by a Bonferroni test. Error bars = SEM; n = 30–40). (D) Most chemosensory and osmosensory-defective mutants that possess strong or weak cilium-structure defects based on a dye-filling assay ([46]; shown in blue and green, respectively) have exploration defects. Chemosensory-defective mutants *odr-1*, *odr-3*, *odr-8*, *odr-10*, *tax-2*, and *tax-4*; osmosensory-defective mutants *osm-7* and *osm-10*; and thermosensory-defective mutant *ttx-1(p767)*, also have exploration defects. In contrast, chemosensory mutants *che-1*, *che-6*, *che-7*, *odr-2*, *odr-4*, *odr-5*, *odr-7*, osmosensory mutants *osm-9* and *osm-11*, oxygen-sensing mutant *gcy-35*, and mechanosensory-defective mutants *mec-3*, *mec-4*, and *glr-1* do not have exploration defects. The canonical null allele *ocr-2(ak47)* (strain VM396; indicated with an asterisk) showed a significant exploration defect that was not observed with the presumably null allele *ocr-2 (ok1711)*. Outcrossed strain BJH2763 *ocr-2(ak47)* did not show a similar exploration defect, suggesting that the *ocr-2 (ak47)* mutation per se does not cause exploration defects (S3A Fig; see text for details). Shown is the number of squares (out of 50) single animals covered in ~20 h, normalized to WT. (\*\*\*, P<0.001. \*\*, P<0.01. \*, P<0.05. ns, P>0.05. One way ANOVA followed by a Bonferroni test. Error bars = SEM; n = 10–35).

exploration defects. We conclude that ciliated neurons that are involved in chemosensation and osmosensation play a significant role in the exploration behavior.

## GRK-2 controls movement quiescence in an opposite way to EGL-4

*C. elegans* locomotory behavior in the presence of food alternates between two discrete states called roaming and dwelling, where the animal either covers long distances or explores locally, respectively [2,4]. The cGMP-dependent protein kinase EGL-4 acts in the sensory neurons to promote dwelling [2]. *egl-4* loss of function mutant animals explore more than wild type

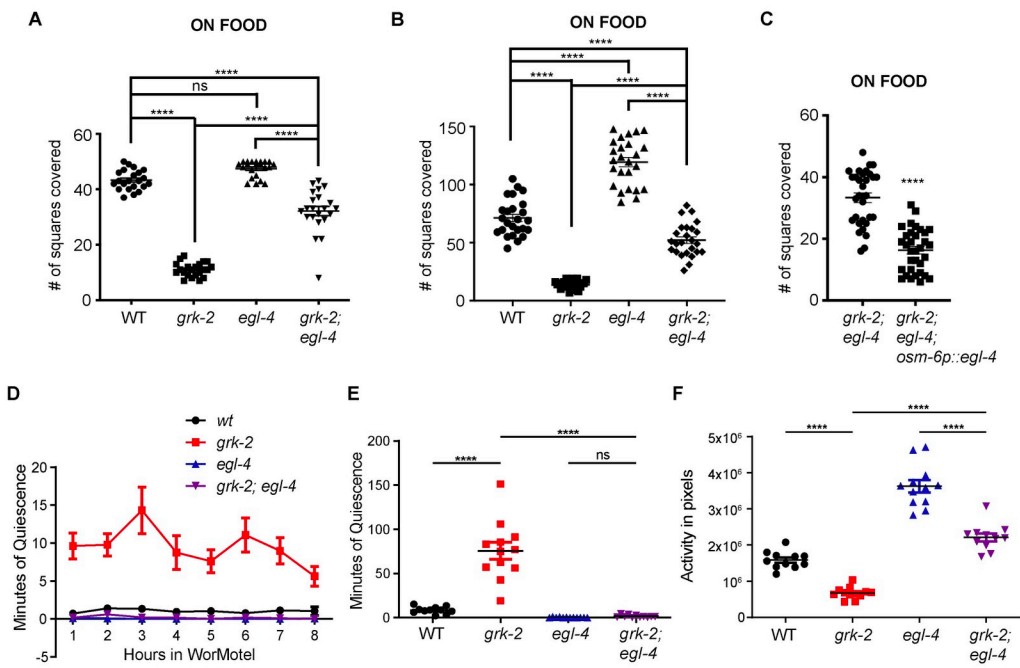

**Fig 4. *egl-4* suppresses the exploration defect, movement quiescence, and reduced activity of *grk-2* mutant animals.**
(A), (B) The *egl-4(n479)* mutation partially suppresses the exploration defect of *grk-2(gk268)* mutant as assayed on 6 cm
(A) and 10 cm (B) plates. Shown is the number of squares (out of 50, (A) or 150, (B)) single animals covered in ~20 h.
Please note that *egl-4* mutant animals explore more than wild type animals as assayed on 10 cm plates. (****, P<0.0001.
ns, P>0.05. One way ANOVA followed by a Bonferroni test. Error bars = SEM; n = 22–26). (C) Expression of *egl-4*
cDNA in ciliated sensory neurons under the *osm-6* promoter reverses the *egl-4* suppression of the exploration defect of
*grk-2* mutant (*osm-6p::egl-4*, transgene *yakEx221*). Shown is the number of squares single animals covered in ~20 h (****,
P<0.0001. ns, P>0.05. Student's t-test. Error bars = SEM; n = 31–33). (D) *grk-2(gk268)* mutant animals show enhanced
movement quiescence, and the *egl-4(n479)* mutation suppresses this phenotype. The graph shows the average minutes
animals spent in quiescence per hour in the WorMotel. Quiescence is the time when zero worm pixels moved between
two frames separated by 10 sec (See Methods). One WT animal was considered an outlier and was not included in the
graph since its values were more than 3 standard deviations from the mean. (E) The graph shows the total minutes
individual animals spent in quiescence in 8 h in the WorMotel. (****, P<0.0001. ns, P>0.05. One way ANOVA. Error
bars = SEM; n = 11–12). (F) *grk-2(gk268)* mutant animals show reduced activity, and the *egl-4(n479)* mutation partially
suppresses this phenotype. The graph shows the activity in pixels of individual animals in 8 h in the WorMotel. Activity is
the number of pixels moved between frames separated by 10 sec. (****, P<0.0001. One way ANOVA. Error bars = SEM;
n = 11–12).

animals [2,4]. Also, *egl-4* mutations suppress the reduced exploration defect of *che-2* mutants
[2]. Thus, we examined whether *egl-4(n479)* mutant animals suppress the *grk-2* exploration
defect. We noticed that when the exploration assay was performed on 6 cm plates, *egl-4* mutant
animals didn't seem to explore significantly more than wild type animals, probably due to a
ceiling effect (Fig 4A). In contrast, when the assay was performed on 10 cm plates, *egl-4* mutant
animals explored significantly more than wild type animals (Fig 4B). In both cases, the *egl-4
(n479)* mutation significantly suppressed the *grk-2* exploration defect (Fig 4A and 4B). Expression of EGL-4 in ciliated neurons reversed the *egl-4* suppression of the *grk-2* exploration defect
(Fig 4C), suggesting that GRK-2 and EGL-4 act in opposite ways in the ciliated neurons to
affect the exploration behavior.

EGL-4 also functions in sensory neurons to promote a *C. elegans* sleep state during lethargus (lethargus quiescence; [3]). The described role of EGL-4 in quiescence prompted us to
examine whether the exploration defect of *grk-2* is the result of movement quiescence. We
measured activity and quiescence in *C. elegans* larval stage 4 (L4) animals by imaging

individual animals cultivated in a WorMotel, which is a micro-fabricated device with multiple agar-filled wells [47,48]. Single L4 animals were placed in WorMotel, imaged for 8 h, and their quiescence and activity were quantified. *grk-2(gk268)* mutant animals showed reduced body movement activity and increased quiescence in this 8 h period and the *egl-4(n479)* mutation suppressed these phenotypes (Fig 4D–4F). We conclude that *grk-2* mutant animals have enhanced movement quiescence that is suppressed by *egl-4*, indicating that *grk-2* and *egl-4* act in opposite directions to regulate movement quiescence. These results also suggest that part of the exploration defect of *grk-2* mutant might be the result of increased movement quiescence.

In addition to cGMP signaling through EGL-4, cAMP signaling pathways act through the salt-inducible kinase KIN-29 in sensory neurons to regulate sensory behaviors [49]. *kin-29* plays an important role in the metabolic regulation of sleep [50] and *kin-29* mutants, like *egl-4* mutants, have reduced sleep. However, in contrast to *egl-4* mutant animals, *kin-29(oy38)* mutant animals did not explore more than wild-type animals and the *kin-29(oy38)* mutation did not rescue the exploration defect of *grk-2(gk268)* mutant (S3B Fig). We conclude that KIN-29 does not play a role in the GRK-2 pathway to control exploration and movement quiescence.

## APTF-1 promotes GRK-2-modulated movement quiescence and exploration

To further explore the role of sleep in the reduced exploration of *grk-2* mutant, we manipulated sleep in *grk-2* mutant animals. Sleep behavior is controlled by the AP2 transcription factor APTF-1, which is required for the development of the sleep-promoting RIS interneuron [51]. Although the exploration behavior of *aptf-1(gk794)* mutant animals was similar to that of wild type animals, the *aptf-1* mutation partially suppressed the decreased exploration phenotype of *grk-2* mutant animals (Fig 5A). Similarly, *aptf-1* mutant partially suppressed both the enhanced quiescence and reduced activity phenotypes of *grk-2* mutant (Fig 5B–5D). We conclude that increased sleep is partially responsible for the reduced exploration phenotype of *grk-2* mutant animals.

APTF-1 promotes sleep by turning on the expression of the sleep-inducing neuropeptide FLP-11 in the RIS neuron. At sleep onset, increases in RIS calcium activity cause the release of FLP-11, which induces quiescence [9]. *flp-11(tm2706)* mutant did not significantly affect the exploration or quiescence phenotypes of *grk-2* mutant (S4A–S4D Fig), suggesting that GRK-2 acts independently of FLP-11 in regulating quiescence, and that the role of APTF-1 in adult movement quiescence is partially independent of FLP-11. Alternatively, since the effect of *flp-11* on sleep is weaker than that of *aptf-1* [9], our assays might not be sensitive enough to detect a phenotype for *flp-11*.

*C. elegans* displays an additional sleep-like quiescent behavior which occurs in response to exposure to cellular stressors (stress-induced quiescence). Mutants lacking the transcription factor CEH-17 lack a functional ALA neuron, a neuron which promotes stress-induced quiescenc [13,14,52,53]. In contrast to the *aptf-1* mutation, the *ceh-17(np1)* mutation did not affect the exploration or quiescence phenotypes of *grk-2* (S4E–S4H Fig), suggesting that GRK-2 acts independently of the ALA neuron to control movement quiescence.

## NPR-1 negatively controls GRK-2- modulated exploration behavior

The *C. elegans* neuropeptide receptor NPR-1 negatively regulates arousal by inhibiting the activity of a sensory circuit that is connected by gap junctions to the RMG interneurons [54]. *npr-1* loss-of-function mutant animals have increased locomotion both in lethargus and in the adult stage due to heightened activity of the RMG circuit [8,30–32,54]. *npr-1(ky13)* mutant animals explored more than wild type animals and *npr-1(ky13)* suppressed the exploration

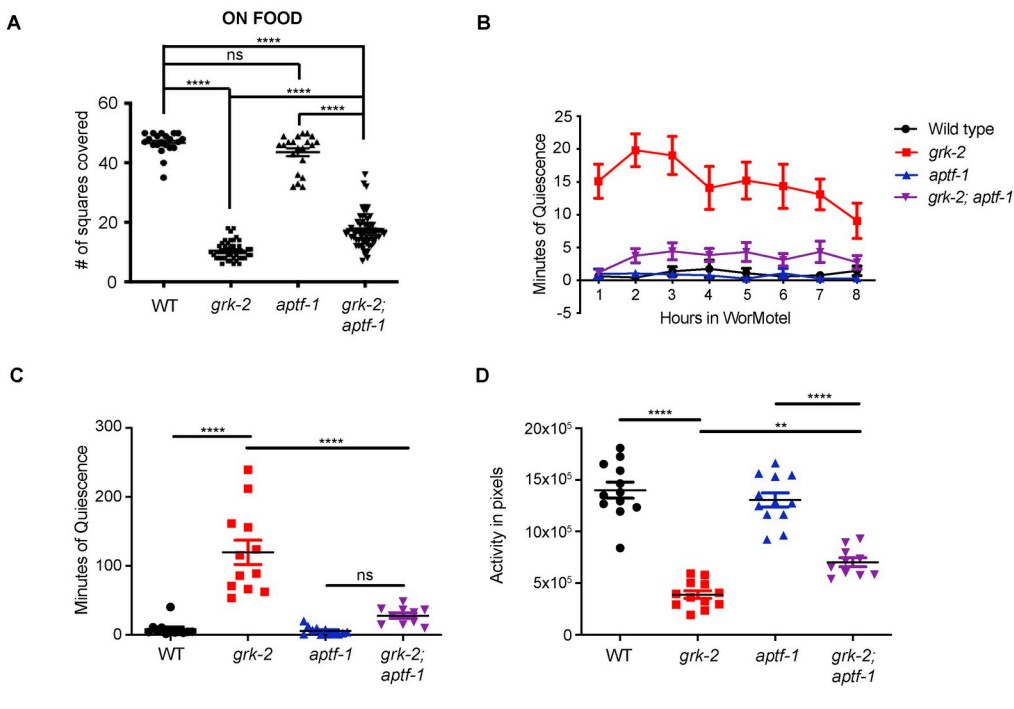

**Fig 5. *aptf-1* partially suppresses the exploration defect, movement quiescence, and reduced activity of *grk-2*.** (A) The *aptf-1(gk794)* mutation partially suppresses the exploration defect of *grk-2(gk268)* mutant. Shown is the number of squares single animals covered in ~20 h. (****, P<0.0001. ns, P>0.05. One way ANOVA followed by a Bonferroni test. Error bars = SEM; n = 21–62). (B) *grk-2(gk268)* mutant animals show enhanced movement quiescence, and the *aptf-1 (gk794)* mutation partially suppresses this phenotype. The graph shows the average minutes animals spent in quiescence per hour in the WorMotel. Quiescence is the time when zero pixels moved between two frames separated by 10 sec (See Methods). (C) *grk-2(gk268)* mutant animals show enhanced movement quiescence, and the *aptf-1(gk794)* mutation partially suppresses this phenotype. The graph shows the total minutes individual animals spent in quiescence in 8 h in the WorMotel. (****, P<0.0001. ns, P>0.05. One way ANOVA. Error bars = SEM; n = 10–12). (D) *grk-2(gk268)* mutant animals show reduced activity, and the *aptf-1(gk794)* mutation partially suppresses this phenotype. The graph shows the activity in pixels of individual animals in 8 h in the WorMotel. Activity is the number of pixels moved between frames separated by 10 sec. (****, P<0.05. **, P<0.01. One way ANOVA. Error bars = SEM; n = 10–12).

defect of *grk-2(gk268)* (Fig 6A and 6B). Moreover, *npr-1(ky13)* also suppressed the *grk-2* dispersal defect in the absence of food (Fig 6C). These results suggest that *npr-1* and *grk-2* act in opposite ways to affect the exploration behavior.

The FMRFamide-related neuropeptides (FaRPs) FLP-18 and FLP-21 are known NPR-1 ligands [55,56]. While *flp-18(db99)* and *flp-21(ok889)* single mutants did not have an exploration defect, a *flp-18* mutation partially suppressed the defective exploration phenotype of *grk-2* (Fig 6D). The *flp-21* mutation did not affect the *grk-2* exploration defect and *flp-21; flp-18* mutants were not stronger suppressors than *flp-18* mutants, suggesting that the *flp-21(ok889)* mutation does not suppress the *grk-2* exploration defect (Fig 6E). The NPR-1 ligand FLP-18 only having a partial effect suggests that it is not the only ligand for NPR-1 in this pathway. Alternatively, NPR-1 may have ligand-independent activity.

NPR-1 is expressed in several neurons including neurons of the RMG circuit (ASH, URX, RMG) [57] and expression in the RMG circuit completely rescues the increased locomotion of *npr-1* mutants during lethargus [8,31]. Moreover, expression solely in the RMG interneuron rescues the enhanced locomotion speed of *npr-1* mutants [54] while NPR-1 acts in the ASH sensory neurons to inhibit sensory responses [8]. To examine whether ASH-specific

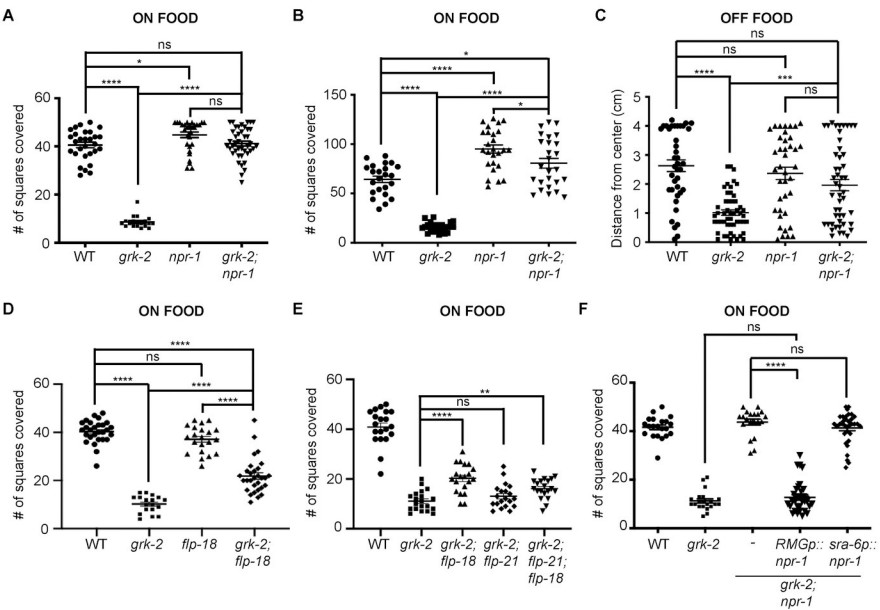

**Fig 6.** ***npr-1* suppresses the exploration defect of *grk-2* mutant animals.** (A) The *npr-1(ky13)* mutation suppresses the exploration defect of *grk-2(gk268)* mutants as assayed on 6 cm plates. Shown is the number of squares (out of 50) single animals covered in ~20 h. (****, P<0.0001. *, P<0.05. ns, P>0.05. One way ANOVA followed by a Bonferroni test. Error bars = SEM; n = 20–42). (B) The *npr-1(ky13)* mutation suppresses the exploration defect of *grk-2(gk268)* mutant as assayed on 10 cm plates. Shown is the number of squares (out of 150) single animals covered in ~20 h. (****, P<0.0001. *, P<0.05. One way ANOVA followed by a Bonferroni test. Error bars = SEM; n = 23–25). (C) The *npr-1 (ky13)* mutation suppresses the dispersal defect of *grk-2(gk268)* in the absence of food. (****, P<0.0001. ***, P<0.001. ns, P>0.05. One way ANOVA followed by a Bonferroni test. Error bars = SEM; n = 39–51). (D) The *flp-18(db99)* mutation partially but significantly suppresses the exploration defect of *grk-2(gk268)*. (****, P<0.0001. ns, P>0.05. One way ANOVA followed by a Bonferroni test. Error bars = SEM; n = 20–30). The data points shown for WT and *grk-2* mutant are the same as in S4A Fig (these experiments were run in parallel). (E) The *flp-21(ok889)* mutation does not suppress the exploration defect of *grk-2(gk268)*. (****, P<0.0001. **, P<0.01. ns, P>0.05. One way ANOVA followed by a Bonferroni test. Error bars = SEM; n = 20–26). (F) Expression of *ncs-1p::nCre* and *flp-21p*::LoxStopLox::NPR-1 (expression in RMG neurons (54); *RMGp::npr-1*, transgene *yakEx206*) reverses the *npr-1* suppression of the exploration defect of *grk-2(gk268)* mutant. Expression of *npr-1* cDNA under the *sra-6* promoter (expression in ASH and ASI neurons; *sra-6p::npr-1*, transgene *yakEx209*) fails to reverse the *npr-1* suppression of the exploration defect of *grk-2(gk268)* (****, P<0.0001. ns, P>0.05. One way ANOVA followed by a Bonferroni test. Error bars = SEM; n = 21–33).

expression of *npr-1* is sufficient to reverse the *npr-1* suppression of the exploration defect of *grk-2* mutants we expressed the *npr-1* cDNA under the control of the ASH (and ASI)—specific *sra-6* promoter (*sra-6p*; [40]). Expression of *npr-1* under the *sra-6* promoter did not reverse the *npr-1(ky13)* suppression of the exploration defect of *grk-2* (Fig 6F).

We next addressed whether RMG- specific expression of *npr-1* is sufficient to reverse the *npr-1*- suppression of the exploration defect of *grk-2*. Since RMG- specific promoters are not known, we used a described intersection strategy to drive *npr-1* expression only in the RMG neuron [54]. In this approach, we used the *ncs-1* promoter to drive the Cre recombinase and the *flp-21* promoter to drive *npr-1* preceded by a floxed STOP site. [54]. Expression of *npr-1* by this method in *grk-2(gk268); npr-1(ky13)* mutant animals fully reversed the *npr-1* suppression of the exploration defect of *grk-2* (Fig 6F). *flp-21* and *ncs-1* are coexpressed in several neurons (including the RMG neurons), but of these neurons, *npr-1* is present only in ASE, ASG, and RMG. Since ASE and ASG are gustatory neurons that have not been implicated in NPR-1 function in locomotion, our experiments suggest that NPR-1 functions in RMG to negatively control the GRK-2 pathway that controls the exploration behavior.

## FLP-1 negatively controls GRK-2- mediated exploration behavior

In a literature search for additional neuropeptides that could play a role in GRK-2- modulated exploration, we came across FLP-1. *flp-1(ok2811)* null mutant animals are hyperactive and have a loopy locomotion posture characterized by deep body bends [34–37]. Interestingly, although *flp-1(ok2811)* mutant animals explored as well as WT animals in the presence of food, the *flp-1(ok2811)* mutation significantly suppressed the *grk-2(gk268)* exploration defect (Fig 7A). Moreover, *grk-2(gk268); flp-1(ok2811); flp-18(db99)* triple mutant animals behaved like the *grk-2(gk268); flp-1(ok2811)* double mutant, suggesting that FLP-1 and FLP-18 act in the same pathway to negatively modulate GRK-2-mediated exploration behavior (Fig 7B). Expression of FLP-1 under its own promoter and under the AVK neuron-specific *twk-47* promoter reversed the *flp-1* suppression of the *grk-2* exploration phenotype, suggesting that expression of FLP-1 in AVK interneurons is sufficient to restore its function (Fig 7C). To assess whether *npr-1* and *flp-1* act in the same pathway, we built the triple mutant *grk-2; flp-1; npr-1* and compared its phenotype with that of the *grk-2; npr-1* double mutant. We found that

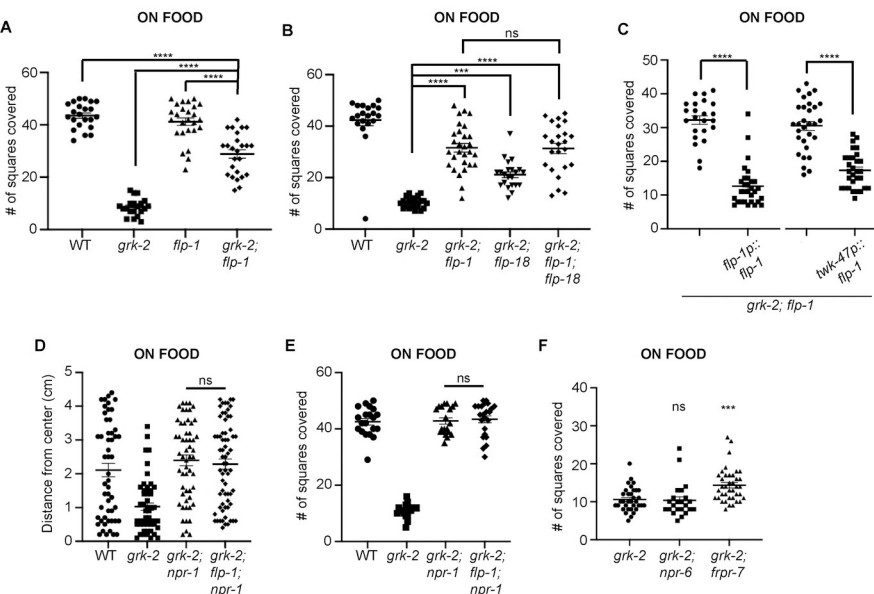

**Fig 7. *flp-1* significantly suppresses the exploration defect of *grk-2*.** (A) *grk-2(gk268)* mutant animals have an exploration defect and *flp-1(ok2811)* significantly suppresses this defect. Shown is the number of squares (out of 50) single animals covered in ~20 h. (****, P<0.0001. One way ANOVA followed by a Bonferroni test. Error bars = SEM; n = 20–26). (B) *flp-1* and *flp-18* act in the same pathway to regulate exploration. *flp-1(ok2811)* and *flp-18(db99)* partially suppress the exploration defect of *grk-2(gk268)* mutant and the *grk-2(gk268); flp-1(ok2811); flp-18(db99)* triple mutant behaves like the *grk-2(gk268); flp-1(ok2811)* double mutant. (****, P<0.0001. ***, P<0.001. ns, P>0.05. One way ANOVA followed by a Bonferroni test. Error bars = SEM; n = 20–28). (C) Expression of *flp-1* under its own promoter (*flp-1p::flp-1*, transgene *yakEx231*) or under the *twk-47* promoter (expression in AVK neurons; *twk-47p::flp-1*, transgene *yakEx216*), reverses the *flp-1* suppression of the exploration defect of *grk-2(gk268)* mutant. (****, P<0.0001. Student's t-test. Error bars = SEM; n = 23–30). (D) *npr-1(ky13)* suppresses the dispersal defect of *grk-2 (gk268)* mutant and the *grk-2(gk268); flp-1(ok2811); npr-1(ky13)* triple mutant behaves like the *grk-2(gk268); npr-1 (ky13)* double mutant. Shown is the distance (cm) animals traveled from the center of the plate in the presence of food in 1 h. (ns, P>0.05. One way ANOVA followed by a Bonferroni test. Error bars = SEM; n = 51–67). (E) *npr-1(ky13)* suppresses the exploration defect of *grk-2(gk268)* mutant and the *grk-2(gk268); flp-1(ok2811); npr-1(ky13)* triple mutant behaves like the *grk-2(gk268); npr-1(ky13)* double mutant. Shown is the number of squares (out of 50) single animals covered in ~20 h (ns, P>0.05. One way ANOVA followed by a Bonferroni test. Error bars = SEM; n = 20–23). (F) *frpr-7(gk463846)* partially but significantly suppresses the exploration defect of *grk-2(gk268)* mutant. *npr-6(tm1497)* has no effect on *grk-2(gk268)* exploration. (***, P<0.001. ns, P>0.05. One way ANOVA followed by a Bonferroni test. Error bars = SEM; n = 23–39).

*grk-2; flp-1; npr-1* behaves like *grk-2; npr-1* mutant animals, suggesting that either NPR-1 and FLP-1 act in the same pathway or that we cannot detect further suppression due to a ceiling effect (Fig 7D and 7E). Previous reports suggest that NPR-6 and FRPR-7 are low- and high-affinity FLP-1 receptors, respectively [36]. We tested the exploration behavior of *grk-2(gk268); npr-6(tm1497)* and *grk-2(gk268); frpr-7(gk463846)* double mutants. *npr-6(tm1497)* did not affect the exploration behavior of *grk-2(gk268)*, but *frpr-7(gk463846)* partially but significantly suppressed the exploration defect of *grk-2(gk268)* (Fig 7F). These results suggest that *frpr-7* but not *npr-6* could be playing a role in the exploration behavior.

## GRK-2 negatively regulates FLP-1 secretion from the AVK neuron

To examine the possibility of a functional connection between GRK-2 and FLP-1, we assessed whether GRK-2 affects the levels of FLP-1 secretion using the coelomocyte uptake assay [58–60]. Proteins that are secreted into the *C. elegans* body cavity are endocytosed by scavenger cells named coelomocytes (Fig 8A). We expressed the fusion protein FLP-1::mCherry in AVK neurons of *flp-1* single mutant animals and of *grk-2; flp-1* double mutant animals and quantified the intensity of fluorescence in coelomocytes as a readout of secreted FLP-1::mCherry (see Methods; [36]). Previous work confirmed that FLP-1::mCherry follows the dense core vesicle route for neuropeptide secretion and used the FLP-1::mCherry secretion assay as a way of evaluating FLP-1 secretion [36]. Notably, FLP-1::mCherry expression in *grk-2; flp-1* reversed the suppression of the exploration defect of *grk-2* by *flp-1*, suggesting that FLP-1 in the FLP-1::mCherry reporter retains its function in exploration (S5 Fig). *grk-2; flp-1* mutants expressing FLP-1::mCherry had significantly increased levels of mCherry in coelomocytes in comparison to *flp-1* mutant animals (Fig 8A and 8B). To examine whether background fluorescence in the coelomocytes of *grk-2; flp-1* mutants also changed, together with FLP-1::mCherry we co-expressed a reporter expressing GFP driven by the coelomocyte-specific promoter *unc-122* and tested the levels of GFP fluorescence in the coelomocytes of *flp-1* and *grk-2; flp-1* mutants. We found that GFP fluorescence remained unchanged in *grk-2; flp-1* mutants suggesting that

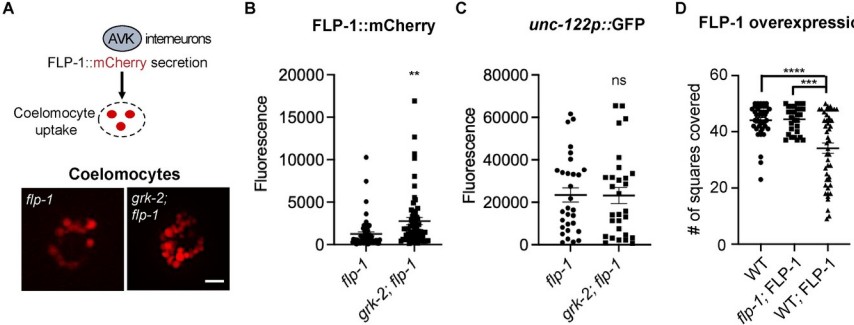

**Fig 8. *grk-2* mutant animals expressing FLP-1::mCherry show increased fluorescence in their coelomocytes.** (A) (Up) Schematic diagram of the coelomocyte uptake assay used to quantify FLP-1 secretion from AVK interneurons. (Down) Representative images of coelomocytes of *flp-1(ok2811)* and *grk-2(gk268); flp-1(ok2811)* mutant animals expressing a rescuing construct of FLP-1::mCherry (FLP-1::mCherry; transgene *yakEx256*). Scale bars = 10 μm. (B) mCherry fluorescence intensity of coelomocytes of *flp-1(ok2811)* and *grk-2(gk268); flp-1(ok2811)* mutant animals expressing FLP-1::mCherry and a coelomocyte marker *unc-122*::GFP. (FLP-1::mCherry; transgene *yakEx256*). (**, P<0.01. Student's t-test. Error bars = SEM; n = 57–59). (C) GFP fluorescence intensity of coelomocytes of *flp-1 (ok2811)* and *grk-2(gk268); flp-1(ok2811)* mutant animals expressing FLP-1::mCherry and a coelomocyte marker *unc-122*::GFP (*unc-122::GFP*; transgene *yakEx256*). (ns, P>0.05. Student's t-test. Error bars = SEM; n = 29–30). (D) Overexpression of FLP-1 reduces the exploration behavior of wild type animals. Shown is the number of squares covered in ~20 h by single wild type animals, *flp-1* mutant expressing low levels of FLP-1::mCherry (transgene *yakEx256*; 5 ng/ul injected), and wild type animals overexpressing FLP-1 (transgene *yakEx261*; 60 ng/ul injected). (****, P<0.0001. ***, P<0.001. One way ANOVA followed by a Bonferroni test. Error bars = SEM; n = 29–47).

the increase in FLP-1::mCherry fluorescence in *grk-2; flp-1* coelomocytes is specific to FLP-1::mCherry (Fig 8C). These results suggest that GRK-2 affects the level of secreted FLP-1, which could explain at least in part the exploration defect observed in mutants lacking GRK-2. Consistent with this idea, multi-copy expression of FLP-1::mCherry under the *flp-1* promoter reduced the exploration behavior of wild type animals (Fig 8D). These results altogether indicate that GRK-2 negatively regulates the amount of FLP-1 that is secreted by AVK neurons, resulting in positive regulation of the exploration behavior.

## Discussion

Locomotion is the result of dynamic interactions between a genetically determined central program and feedback mechanisms that modulate the activity of the central program [61]. Feedback is mainly of sensory and neuromodulatory nature and it plays a major role in adapting the pattern of locomotion to environmental or metabolic changes [1]. Here we have demonstrated that locomotor behavioral strategies in the nematode *Caenorhabditis elegans* are altered through the action of the G protein-coupled receptor kinase-2 (GRK-2) in ciliated sensory neurons and neuropeptide signaling in interneurons. We propose that GRK-2 acts in ciliated sensory neurons to control the exploration behavior and the arousal state of the animal by controlling the level of secretion of the neuropeptide FLP-1.

### GRK-2 signaling in ciliated sensory neurons controls exploration

The sensory system influences the expression of different locomotor states in *C. elegans*, like roaming and dwelling [2]. Roaming is a behavioral exploration strategy where the animal covers long distances, while dwelling is a local search behavior strategy where animals restrict their activity to a small region. Previous studies have shown that mutants with diminished sensory responses to soluble and volatile chemicals are defective in the time they spend roaming or dwelling, suggesting that sensory perception is critical for this behavior [2]. Specifically, a class of mutants that lack a normal sensory cilium structure, including *che-2* [62,63], are unable to explore long distances. In this study we show that *grk-2* mutant animals have similarly severe phenotypes as the cilium structure mutants in that they also show reduced exploration. Our findings suggest that the positive role of GRK-2 in exploration behavior is due to its positive role in chemosensation. We show that expression of GRK-2 in the ciliated sensory neurons of *grk-2* mutant animals rescues their exploration defect, suggesting that GRK-2 acts specifically in these neurons to affect exploration. Our effort to narrow down the specific neurons where GRK-2 acts to mediate this behavior revealed that GRK-2 acts in multiple ciliated neurons to control exploration. Interestingly, although *grk-2* mutant animals have strong chemosensory defects, they do not have severe cilia structural defects like the cilium structure mutants [23,64]. Also, expression of GRK-2 in adult *grk-2* mutant animals can rescue the *grk-2* exploration defect suggesting that GRK-2 plays a non-developmental role in ciliated neurons to promote exploration. Our study of mutants with defects in different kinds of sensory perception showed that the activity of the ciliated neurons that are involved in chemosensation and osmosensation play important roles in the exploration behavior, but neurons involved in mechanosensation or oxygen-sensation do not. These results together suggest that impaired sensory neuron function per se can reduce exploration.

Our results show that *grk-2* and the cilium structure mutant *che-2* act in the same genetic pathway to control exploration, again suggesting that impaired ciliated sensory neuron function disrupts exploration. In agreement, we found that mutations in the gene encoding the cGMP-dependent protein kinase EGL-4 that suppress the altered exploration behavior of *che-2* mutants [2], also suppress the exploration defect of *grk-2* mutant animals. Given that

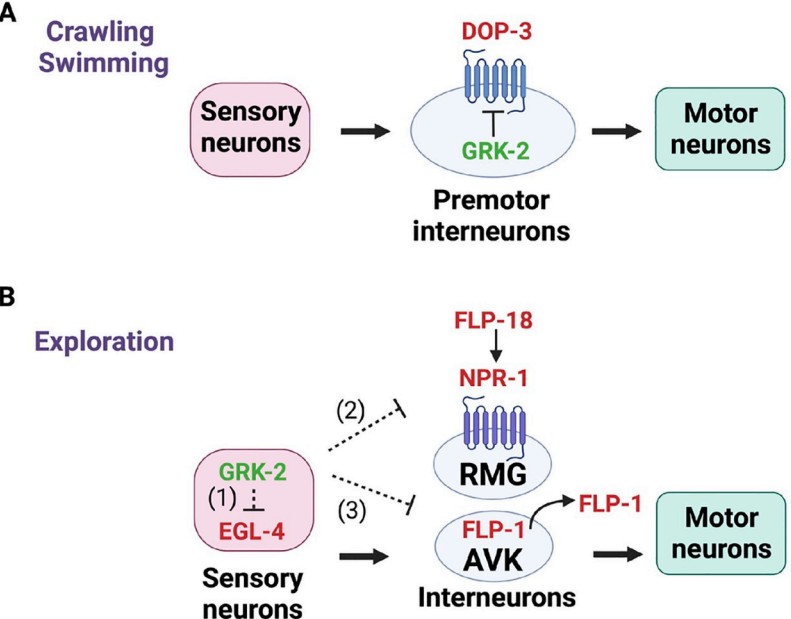

**Fig 9. Model for GRK-2 modulation of locomotion behaviors in *C. elegans*.** GRK-2 positively controls (A) crawling and swimming by acting in premotor interneurons in a manner that depends on the D2-like dopamine receptor DOP-3 [27,28] and (B) exploration by acting in ciliated neurons in an opposite way from EGL-4. We propose a model in which GRK-2 acts in multiple ciliated sensory neurons to positively control sensory perception and (1) inhibit the activity of the EGL-4 kinase in the same ciliated neurons, (2) negatively regulate the neuropeptide receptor NPR-1 in RMG interneurons in a cell-nonautonomous way (e.g. by regulating the level of secretion of an NPR-1 ligand), and (3) negatively control the levels of FLP-1 secretion from AVK interneurons. AVKs play critical roles in regulating locomotion by modulating motor neuron activity [36]. Since the ciliated neurons and AVK interneurons do not have direct connections, we propose that GRK-2 controls FLP-1 secretion from AVK interneurons in a cell-nonautonomous way through a neuromodulatory pathway. Proteins that positively regulate crawling, swimming, or exploration are shown in green and proteins that negatively regulate crawling, swimming, or exploration are shown in red. Dashed lines show indirect interactions. Created with BioRender.com.

expression of EGL-4 in the ciliated neurons reverses this suppression, we propose that both GRK-2 and EGL-4 act in the same ciliated neurons to control exploration but in opposite ways. Given that both GRK-2 and EGL-4 act in multiple ciliated neurons to control exploration behavior and that other chemosensory and osmosensory mutants have similar exploration defects as *grk-2*, we support a model in which GRK-2 acts in ciliated neurons to positively control sensory perception and affect the exploration state of the animal by regulating EGL-4. We propose that in *grk-2* and cilium structure mutants, impaired sensory perception leads to an increase in EGL-4 activity and decreased exploration. A similar model was supported by previous studies, which showed that EGL-4 in sensory neurons controls body size and locomotory states [2].

In previous work we have shown that GRK-2 regulates crawling and swimming [27,28] by negatively regulating the D2-like dopamine receptor DOP-3 in premotor interneurons (Fig 9A). In the case of chemosensation and exploration, the target(s) of GRK-2, which are located in chemosensory neurons and not interneurons, are currently unknown (Fig 9B). Similarly, we do not yet know the identity of the phosphorylation target(s) for EGL-4.

## GRK-2 controls movement quiescence

Animals undergo periods of quiescence and arousal in response to changes in their environment or metabolic state. For example, *C. elegans* increases its exploration behavior when it is

removed from food [10,11] and shows quiescence during larval molting (lethargus) and under starvation, satiety, and stress [3,12–17]. The cGMP-dependent protein kinase EGL-4 acts in sensory neurons to promote *C. elegans* sleep where the animals become quiescent during lethargus [3]. This result, together with the observed opposite functionality between EGL-4 and GRK-2 in ciliated neurons prompted us to examine whether *grk-2* mutants have movement quiescence defects. Indeed, we found that *grk-2* mutant animals have enhanced movement quiescence, which is suppressed by *egl-4*. This result suggests that GRK-2 and EGL-4 act antagonistically to control movement quiescence outside lethargus. We propose that GRK-2 acts in ciliated neurons to control sensory perception and affect movement quiescence by negatively regulating EGL-4. Our findings imply that the exploration defect of *grk-2* and other chemosensory mutants besides *grk-2* might be due in part to enhanced movement quiescence. This series of data also suggests that *C. elegans* locomotion strategies that have been described as exploration, roaming, and arousal might reflect the same behavior.

In addition to EGL-4, the AP2 transcription factor APTF-1 also plays an important role in regulating sleep, largely by regulating expression of the sleep-inducing neuropeptide FLP-11 in the RIS interneuron [51]. Indeed, we found that *aptf-1* mutants partially suppress the decreased exploration phenotype, enhanced quiescence, and reduced activity of *grk-2*. This result suggests that increased sleep is partially responsible for the reduced exploration phenotype of *grk-2* mutant animals. Surprisingly, we found that *flp-11* mutations do not suppress the enhanced quiescence of *grk-2* mutant animals. One possible explanation is that APTF-1 acts through additional targets besides FLP-11, which is also supported by previous studies [9]. Alternatively, FLP-11 might be playing a smaller role than APTF-1 in adult quiescence and our assays might not be sensitive enough to detect it.

## Neuromodulation through FLP-1 negatively regulates exploration

Mutants lacking the neuropeptide receptor NPR-1 have been described as a model for arousal both during lethargus and in adults [8,31–33]. NPR-1 is proposed to act in the RMG interneuron to inhibit the activity of a sensory circuit that is connected to RMG via gap junctions (54). In *npr-1* mutants, enhanced sensory activity causes aroused locomotion [8,31–33,65]. Our results show that *npr-1* mutants suppress the exploration defects of *grk-2* mutant animals, and that NPR-1 acts in RMG neurons to mediate this effect. The fact that *npr-1* is epistatic to *grk-2* suggests that GRK-2 negatively regulates NPR-1 –in a *grk-2* mutant, increase in NPR-1 activity inhibits exploration (Fig 9B). Since GRK-2 acts in ciliated neurons and NPR-1 acts in RMG neurons to control exploration, this suggests that GRK-2 activity regulates NPR-1 in a cell-nonautonomous way. Of the reported NPR-1 ligands FLP-18 and FLP-21, we found that only FLP-18 acts in the GRK-2 pathway that mediates exploration, since *flp-18* mutants partially suppress the exploration defect of *grk-2* [55,56]. Interestingly, we also found that mutants in the gene encoding the neuropeptide FLP-1 partially suppress the *grk-2* exploration defect and that FLP-1 acts in the same genetic pathway as FLP-18. FLP-1 has not been reported to be a ligand for *C. elegans* NPR-1, but it was shown to activate the *Girardia tigrina* neuropeptide receptor 1 GtNPR-1 when expressed in Chinese hamster ovary cells [66]. This finding together with our genetic observations could suggest that FLP-1 directly acts through NPR-1 to affect exploration. Alternatively, and since *flp-1* and *flp-18* mutants do not act in an additive way to suppress the *grk-2* exploration defect, FLP-1 could be acting in the GRK-2 pathway through a different receptor than NPR-1. The partial suppression of the *grk-2* exploration defect by *frpr-7*, suggests that the high-affinity FLP-1 receptor FRPR-7 could be playing a role.

Notably, Oranth et al (2018) [36] suggested that after >10 min off food, AVK releases FLP-1 peptides to promote dispersal. Our genetic experiments show that mutants in *flp-1* suppress

the exploration defect of *grk-2* mutant animals and that overexpression of FLP-1 reduces exploration in the presence of food. This apparent discrepancy might be attributed to different FLP-1 functionality in the presence or absence of food or when chemosensation is impaired. Our genetic results together with our finding that GRK-2 modulates the secretion levels of FLP-1 suggest that GRK-2 negatively regulates FLP-1 secretion, either directly or indirectly (Fig 9B). Since GRK-2 acts in ciliated neurons to affect exploration and GRK-2 modulates FLP-1 secretion from AVK neurons we propose that GRK-2 acts in ciliated neurons to mediate FLP-1 secretion from AVKs in a cell-nonautonomous way. Given that there is no known synaptic input from chemosensory neurons to AVKs [67] we suggest that GRK-2-driven regulation of sensory neuron activity affects a neuromodulatory pathway that drives FLP-1 secretion from AVKs (Fig 9B). AVKs were previously reported to play important roles in regulating locomotion by modulating motor neuron activity [36]. Alternatively, GRK-2 action in ciliated neurons may lead to modulation of the neuronal circuit downstream of the ciliated neurons that eventually affects secretion of FLP-1. According to our model, GRK-2 activity positively regulates chemosensory neuron activity, which in turn inhibits EGL-4 activity, modulating exploration and the arousal state of *C. elegans* through neuromodulatory signaling acting in interneurons (Fig 9B). Thus, an active chemosensory system can guide *C. elegans* towards exploration, while a defective chemosensory system inhibits dispersal and puts the animal in a state of quiescence.

## Methods

### Strains

Strains were maintained at room temperature or 20˚ on the OP50 strain of *E. coli* [68]. The Supplementary Information contains a list of strains used, including their full genotypes (S1 Table; List of strains).

### Constructs and transgenes

The Supplementary Information contains a list of constructs used (S2 Table; List of plasmids). All constructs made in this study were constructed using the Gateway system (Invitrogen): a promoter region, a gene region (cDNA) or fluorescent tag, and a N- or C-terminal 3'UTR or fluorescent tag fused to a 3'UTR were cloned into the destination vector pCFJ150 [69]. For the cell-specific rescuing experiments, an operon GFP was included in the expression constructs downstream of the 3'UTR [70]. This resulted in expression of untagged protein, but also allowed for confirmation of proper promoter expression by monitoring GFP expression.

Extrachromosomal arrays were made by standard injection and transformation methods [71]. We injected 5–10 ng/ul of the expression plasmid for *grk-2* and *flp-1* rescue experiments and 20 ng/ul of the plasmids for other experiments. For the *yakEx261* FLP-1 overexpression experiment in Fig 8D we injected 60 ng/ul. We injected 50 ng/ul of the *unc-122*::GFP co-injection marker, 1 ng/ul of *myo-2p*::mCherry, 5 ng/ul of *myo-3p*::mCherry, and 5 ng/ul of *rab-3p*::mCherry. We isolated two to three independent lines. The results shown in the figures are for two of the lines tested.

### Exploration assays

The exploration assay is an established assay that measures the ability of the animals to explore (wander) during long periods of time [4]. Exploration assays were performed on 6 or 10 cm nematode growth medium (NGM) plates uniformly seeded with 200 or 500 ul of the OP50 strain of *E. coli*, respectively, and spread with a glass spreader. 10 cm plates were used in the case of mutants that explore more than wild type animals (e.g. *egl-4*, *npr-1*) in order to avoid a

ceiling effect because wild type animals usually explore most of the area of a 6 cm plate in a 20h period. Bacteria were grown on the NGM agar surface at room temperature for approximately 48 h and the plates were stored at 4˚C until needed. On the day of the experiment, individual first-day adult animals were transferred to the center of each plate and were allowed to explore for 18–20 h. Then the plates were superimposed on a 50- or 150- square grid and we counted the number of squares entered by the worm tracks. Approximately 10–40 animals were tested per strain. Every strain was tested in two or three independent trials. The number "n" in figure legends indicates the total number of animals sampled (animals from two or three different trials are combined).

### Dispersal assays

The dispersal (or radial locomotion) assay is an established assay that measures the distance animals travel from the middle of the plate in a period [38]. Since the dispersal assay measures distance, it is only performed for short periods of time (1h on food or 20 min on plates without food). Dispersal assays were performed on 10 cm nematode growth medium (NGM) plates that were uniformly seeded with 500 μl of the OP50 strain of *E. coli* and spread with a glass spreader. Bacteria were grown at room temperature for approximately 48 h and the plates were stored at 4˚C until needed. Approximately 20 first day adult animals were transferred to the center of a plate and were allowed to explore for 1 h. Then the positions of the worms were marked and the distances of the worms from the starting point were measured. Dispersal assays without food were performed on NGM plates that were not spread with OP50 and the animals were allowed to move freely for 20 min. Animals were first transferred to an intermediate plate without food before being transferred to the assay plate. Approximately 20–60 animals were tested per strain. Every strain was tested in two or three independent trials. The number "n" in figure legends indicates the total number of animals sampled (animals from two to three different trials are combined).

### Heat shock assays

Animals were raised to young adulthood at room temperature (23˚C), then shifted to 34˚C for 2 hr. They were allowed to recover at room temperature (23˚C) for 4 hr and tested.

### Movement quiescence assays

Movement quiescence assays were performed as previously described [48]. Briefly, animals were selected as fourth larval stage L4s the day prior to the assay. They were loaded individually onto the surface of NGM agar-filled wells on a PDMS WorMotel chip. The agar surface in each well was covered in a thin lawn of DA837 *E. coli*. The loaded chip was placed in a 10 cm petri dish lined with damp kimwipes to prevent desiccation and placed under a camera. Images were acquired every 10 seconds for eight hours. The images were then analyzed using a custom written program in Matlab. For more details on this method see [48]. 8–12 animals were tested per condition and compared to 8–12 control animals imaged in the same WorMotel. The "n" in figure legends indicates the total number of animals sampled.

### Body bend locomotion assays

First-day adults were picked to a three-day-old lawn of OP50 and stimulated by poking the tail of the animal with a worm pick. Body bends were then immediately counted for one minute. A body bend was defined as the movement of the worm from maximum to minimum amplitude of the sine wave. Approximately 10 animals were tested per strain. Every strain was tested

on two independent trials. The number "n" in figure legends indicates the total number of animals sampled (animals from the different trials are combined).

## Imaging

For fluorescence imaging, first-day adult animals were mounted on 2% agarose pads and anesthetized with 50 mM sodium azide for ten minutes before placing the cover slip. The animals were imaged using an Olympus FLUOVIEW FV1200 confocal microscope with a 60× UPlanSApo oil objective (NA = 1.35) or a NikonTi2-E Crest X-light V2 spinning disk confocal with a 60x oil objective (NA = 1.4).

## Coelomocyte assays

For coelomocyte imaging we used FLP-1::mCherry (a gift from Alexander Gottschalk; [36]). We imaged the anterior coelomocytes in day 1 adults using a NikonTi2-E Crest X-light V2 spinning disk confocal with a 60x oil 1.4 NA objective. Maximum intensity projections of image stacks were obtained using FluoView software (Nikon). For quantification, the maximum fluorescence intensity was calculated using Fiji and the background fluorescence intensity was subtracted to obtain coelomocyte fluorescence.

## Statistics

Statistical analysis and graphing were carried out using Prism 8 or Prism 9 (GraphPad Software). Normally distributed data sets requiring multiple comparisons were analyzed by a one-way ANOVA or by a one-way ANOVA followed by a Bonferroni test. Normally distributed pairwise data comparisons were analyzed by two-tailed unpaired t tests. $p < 0.05$ (following Bonferroni correction, when appropriate) was considered to be statistically significant (*$p < 0.05$, **$p < 0.01$, ***$p < 0.001$, ****$p<0.0001$). Data are presented as the mean ± standard error (SEM).

## Raw data

S3 Table contains all the data points used for the main and supplementary figures.

## Supporting information

**S1 Fig. *grk-2* acts in parallel to *pdfr-1* and *tph-1* to control exploration.** (A) *grk-2(gk268)* and *pdfr-1(ok3425)* mutants have exploration defects. The double mutant *grk-2(gk268); pdfr-1 (ok3425)* has a stronger phenotype than the single mutants suggesting that these genes act in parallel to control exploration behavior. Shown is the number of squares (out of 50) single animals covered in ~20 h. (****, $P<0.0001$. **, $P<0.01$. One way ANOVA followed by a Bonferroni test. Error bars = SEM; n = 23–28). (B) *grk-2(gk268)* mutant animals explore less and *tph-1(mg280)* more than WT animals. The double mutant *tph-1(mg280); grk-2(gk268)* has an intermediate phenotype suggesting that these genes act in parallel pathways to control exploration behavior. (****, $P<0.0001$. ***, $P<0.001$. One way ANOVA followed by a Bonferroni test. Error bars = SEM; n = 23–41).
(PDF)

**S2 Fig. *grk-2* acts in command interneurons to regulate crawling.** *grk-2* cDNA expression in command interneurons (*Pnmr-1*, transgene *yakSi32*) is sufficient to rescue the slow locomotion of *grk-2(gk268)* mutant animals. (****, $P<0.0001$. ns, $P>0.05$. One way ANOVA followed by a Bonferroni test. Error bars = SEM; n = 20).
(PDF)

**S3 Fig.** *ocr-2(ak47)*, *ocr-2(ok1711,)* **and** *kin-29(oy38)* **mutants do not have exploration defects.** (A) *ocr-2(ak47)*, *ocr-2(ok1717)* (strains BJH2763 and BJH2762, respectively) and *kin-29(oy38)* mutants do not have exploration defects. Shown is the number of squares (out of 50) single animals covered in ~20 h. (ns, P>0.05. One way ANOVA followed by a Bonferroni test. Error bars = SEM; n = 20–24). (B) *kin-29(oy38)* mutant animals do not have an exploration defect and the double mutant *grk-2(gk268); kin-29(oy38)* has an exploration defect similar to *grk-2(gk268)*. (ns, P>0.05. One way ANOVA followed by a Bonferroni test. Error bars = SEM; n = 20–21).
(PDF)

**S4 Fig. Mutations in** *flp-11* **and** *ceh-17* **do not affect the exploration defect and movement quiescence of** *grk-2*. (A) The *flp-11(tm2706)* mutation does not affect the exploration defect of *grk-2(gk268)* mutant animals. Shown is the number of squares (out of 50) single animals covered in ~20 h. (****, P<0.0001. ns, P>0.05. One way ANOVA followed by a Bonferroni test. Error bars = SEM; n = 20–28). The data points shown for WT and *grk-2* mutant are the same as in Fig 6D (these experiments were run in parallel). (B) *grk-2(gk268)* mutant animals show enhanced movement quiescence and the *flp-11(tm2706)* mutation does not affect this phenotype. The graph shows the average minutes animals spent in quiescence per hour in the WorMotel. Quiescence is the time when there were zero pixels moved between two frames separated by 10 sec (See Methods). (C) The graph shows the total minutes individual animals spent in quiescence in 8h in the WorMotel. (****, P<0.0001. ns, P>0.05. One way ANOVA. Error bars = SEM; n = 11–12). (D) *grk-2(gk268)* mutant animals show reduced activity and the *flp-11(tm2706)* mutation does not affect this phenotype. The graph shows the activity in pixels of individual animals in 8 h in the WorMotel. Activity is the number of pixels moved between frames separated by 10 sec. (****, P<0.0001. ns, P>0.05. One way ANOVA. Error bars = SEM; n = 11–12). (E) The *ceh-17(np1)* mutation does not affect the exploration defect of *grk-2(gk268)* mutant animals. Shown is the number of squares (out of 50) single animals covered in ~20 h. (ns, P>0.05. One way ANOVA. Error bars = SEM; n = 20–24). (F) *grk-2(gk268)* mutant animals show enhanced movement quiescence, and the *ceh-17(np1)* mutation does not affect this phenotype. (G) The graph shows the total minutes individual animals spent in quiescence in 8 h in the WorMotel. (*, P<0.05. ***, P<0.001. ns, P>0.05. One way ANOVA. Error bars = SEM; n = 8–12). (H) The graph shows the activity in pixels of individual animals in 8 h in the WorMotel. (****, P<0.0001. ns, P>0.05. Error bars = SEM; n = 8–12).
(PDF)

**S5 Fig. Expression of FLP-1::mCherry in** *grk-2; flp-1* **mutant animals reverses the suppression of the** *grk-2* **exploration defect by** *flp-1*. Shown is the number of squares (out of 50) single animals covered in ~20 h. (****, P<0.0001. ns, P>0.05. One way ANOVA followed by a Bonferroni test. Error bars = SEM; n = 34–39).
(PDF)

**S1 Table. List of strains.**
(DOCX)

**S2 Table. List of plasmids.**
(DOCX)

**S3 Table. Raw data.**
(XLSX)

## Acknowledgments

We dedicate this paper to the memory of Giannis Topalidis, whose support was instrumental for the completion of this manuscript. We thank Alexander Gottschalk for the FLP-1::mCherry plasmid, Ithai Rabinowitch for the *str-1*, *str-2*, and *odr-10* promoter plasmids, and Cori Bargmann for the pEM01 and pEM03 plasmids. Some strains were provided by the CGC, which is funded by the NIH Office of Research Infrastructure Programs (P40 OD010440).

## Author Contributions

**Conceptualization:** Kristen Davis, Jihong Bai, David M. Raizen, Michael Ailion, Irini Topalidou.

**Data curation:** Kristen Davis, Christo Mitchell, Olivia Weissenfels, Irini Topalidou.

**Formal analysis:** Kristen Davis, Christo Mitchell, Olivia Weissenfels, Irini Topalidou.

**Funding acquisition:** Jihong Bai, David M. Raizen, Michael Ailion.

**Investigation:** Kristen Davis, Jihong Bai, David M. Raizen, Michael Ailion, Irini Topalidou.

**Methodology:** Kristen Davis, Irini Topalidou.

**Resources:** David M. Raizen, Michael Ailion.

**Supervision:** David M. Raizen, Irini Topalidou.

**Validation:** Kristen Davis, Christo Mitchell, Olivia Weissenfels, Irini Topalidou.

**Writing – original draft:** Irini Topalidou.

**Writing – review & editing:** Kristen Davis, Jihong Bai, David M. Raizen, Michael Ailion.

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
