## [Decision Letter · Decision Letter 0]

31 Aug 2022

Dear Dr. Topalidou,

Thank you very much for submitting your Research Article entitled 'G Protein-Coupled Receptor Kinase-2 (GRK-2) regulates exploration through neuropeptide signaling in Caenorhabditis elegans' to PLOS Genetics.

The manuscript was fully evaluated at the editorial level and by independent peer reviewers. The reviewers appreciated the attention to an important topic but identified some concerns that we ask you address in a revised manuscript.

We therefore ask you to modify the manuscript according to the review recommendations. Your revisions should address the specific points made by each reviewer.  Please pay particular attention to the the reviewer points about a potential molecular model.  For example, two reviewers raised questions about the grk-2 and egl-4 interaction.  Improvements to Figure 9 could also be helpful.

[LINK]

Yours sincerely,

Denise M. Ferkey

Guest Editor

PLOS Genetics

Gregory P. Copenhaver

Editor-in-Chief

PLOS Genetics

Reviewer's Responses to Questions

**Comments to the Authors:**

Reviewer #1: In this manuscript, Davis et al find that GRK-2, G protein-coupled receptor kinase-2, regulates exploration behavior in the nematode C. elegans. Using genetics, several nicely quantified behavioral assays, cell-specific rescue and imaging, they show that GRK-2 functions in several sensory neurons, opposite of the cGMP-dependent protein kinase EGL-4, the neuropeptide receptor NPR-1 and the neuropeptides FLP-1 and FLP-18. Their work is very important for understanding C. elegans behavior, where many chemotaxis or learning and memory behavioral assays use a readout based on locomotion and are thus affected by the arousal state of the animal. In addition, as all the proteins involved are strongly conserved, similar pathways might be involved in the regulation of locomotion behavior in other animals.

The manuscript is well written, data has been clearly presented and experimental methods, including statistical analysis, has been clearly described.

Here are my comments.

Page 7-8, lines 139-154. In this paragraph the authors discuss the results that suggest that GRK-2 acts in a different pathway than serotonin and PDF signaling. However, in the final sentence of the paragraph, the authors conclude that GRK-2 acts in a different pathway. I think this conclusion is too strong and should be rephrased to something like “Our results suggest that GRK-2 acts in a different pathway…”. The same applies to the last paragraph of the results, page 16. Also there the conclusion is too strong and should be rephrased.

On page 9, the authors discuss the cell-specific rescue experiments to find out in which sensory neurons GRK-2 functions to regulate exploration. They show data where grk-2 was expressed in AWA, AWB and AWC-ON, using three different promoters, but they don’t show to what extent each of these promoters rescue the phenotype. I would strongly urge the authors to do these experiments and report on them, because it could reveal whether expression in either of these cells, or only in one particular cell is sufficient for rescue.

The rescue experiments often show only partial rescue. This could be because the construct was not expressed in all cells required for full rescue, but it could also be that the level of expression was either too low or too high. Are there reasons to believe that grk-2 overexpression could cause a defect? Please discuss this possibility.

On page 10 the authors show data that suggest that che-2 acts in the same pathway as grk-2, as double mutants have the same phenotype as the two singles. It would be good to strengthen this finding, as the authors show the results of only one double and one assay. It would be good to also test the che-2 grk-2 double mutants in the off food assay, and if possible it would be nice to test whether the same holds true for another cilium mutant.

On page 11, the authors state that EGL-4 promotes dwelling as loss of function mutants explore more than wt animals. However, I don’t see this effect in the assays that the authors show. Please explain.

Line 253 refers to fig 4D. Should this be fig 4B-D?

Fig 6A and B shows that npr-1 mutants explore more than wt. Fig 6A shows the result of assays on 6 cm plates, whereas 6B shows results of assays done on 10 cm plates. This is not discussed in the text. Please explain why this is done. And it is unclear to me why the 10 cm assays have not been done for other mutants.

Fig. 6E shows that mutation of flp-21 does not suppress the effect of grk-2 loss of function and the authors conclude that flp-21 does not play a role in this phenotype. However, we see quite often that phenotypes are masked by functional redundancy. I was wondering whether the authors considered testing grk-2 flp-18 flp-21 triple mutants. These mutants might reveal a function of flp-21.

Fig 8c shows the results of a control experiment, which is not mentioned or explained in the tex. Please explain this experiment briefly.

Reviewer #2: This paper is a nice genetic dissection of multiple intersecting pathways that regulate locomotion and sleep in C. elegans. There are many parameters regulating these behaviors: three, roaming/exploration, dwelling, and quiescence, were examined. Previous work has shown that the cGMP-dependent protein kinase EGL-4 regulates the time in quiescence, roaming, and dwelling. For instance, activation of EGL-4 in sensory neurons regulates dwelling.

The authors’ previous work showed that the GRK-2 kinase in premotor interneurons promotes crawling and swimming by antagonizing dopamine DOP-3 receptor signaling. In this paper, the authors confirmed previous work that loss of grk-2 resulted in decreased exploration in the presence or absence of food. However, the decreased exploration was independent of DOP-3 receptor signaling, suggesting that a pathway independent of dopamine signaling is activated during exploration. Using cell-specific promoters to drive grk-2, the authors determined that grk-2 function in specific ciliated sensory neurons (AWA, AWB, AWC, ASH, ADF) partially regulates exploration and acts antagonistically to EGL-4 signaling in the same sensory neurons.

Through epistasis analysis, the authors eliminated (e.g., serotonin or PDF signaling) or identified several pathways in which GRK-2 acts. The decreased exploration of grk-2 mutants is linked to increased sleep, which is partially dependent on the activity of the APTF-1 transcription factor in an unknown cell; loss of aptf-1 partially suppressed the exploration defect in grk-2 mutants but some unknown mechanism. Similarly, loss of FLP-18 release through unknown cells (sensory neurons, interneurons?) results in loss of signaling through the NPR-1 receptor in RMG neurons, which partially suppressed the exploration defect in grk-2 mutants.

Loss of flp-1 partially suppressed the exploration defect of grk-2 mutants. Expression of flp-1 in the AVK interneurons rescued these defects in the grk-2; flp-1 mutants, suggesting that release of FLP-1 peptides affects the exploration behavior. In epistasis analysis, FLP-1 acts in the same pathway as NPR-1; the likely FLP-1 receptor is FRPR-7, which is expressed in many neurons and the coelomocytes. Secretion of FLP-1 peptides was monitored by mCherry presence in coelomocytes; however, FRPR-7 is also expressed in coelomocytes, which slightly muddles the uptake experiments. In addition, the mCherry appeared to be in-frame with the last FLP-1 gene peptide PNFLRFG, which would inactivate this peptide because of the large size of the mCherry relative to the peptide. Furthermore, it was unclear whether this transgene rescued the exploration defect of grk-2 mutants and/or whether the mCherry tag affected the way the FLP-1 propeptide is cleaved. These questions should be addressed either in the Results, Discussion, or both.

The Discussion should be better integrated with a molecular model. The first few pages discussed epistatic relationships without mentioning the proteins, whose molecular identities are known. GRK-2, like EGL-4, is a kinase. Its function is to phosphorylate, but the question is what its target is within the same cells as to where EGL-4 is expressed. Presumably, the two kinases are phosphorylating different targets, thereby activating different pathways leading to different exploration results; however, this was not clearly indicated, particularly in Fig. 9, which was confusing. Are the authors proposing that grk-2 and egl-4 are expressed in the same sensory neurons or different sensory neurons? The arrows and inhibitory arrows are pointing to how proteins are interacting. For example, how does GRK-2, a kinase that is expressed in sensory neurons, negatively regulate FLP-1, which is expressed in the AVK interneuron? If the site of action of the gene is known, it should be so represented in Fig. 9.

Overall, this paper assigned different protein activities to different pathways that modulate different behaviors. The genetics was very well done and highlighted interesting aspects of behavior. The paper suffers from not integrating the work better, particularly in the Discussion.

Minor comments:

1. For exploration and dispersal assays, it was unclear how many trials were conducted. The numbers of animals were indicated, but were animals (and how many animals) tested on one day, multiple days (e.g., multiple trials), etc. The graphs all showed many points on the assays, but do these points represent different trials or different animals in the assays. This information should be explicitly indicated in either the Material and Methods or the figure legends (better option), and the number of trials that were performed and whether animals from different trials were combined in the graphs should be indicated.

2. The specific statistical test should be indicated in each figure legend. The reader should not have to guess.

3. Fig. 1 legend, line 761: Panel B shows the dispersal defect in the presence, not absence, of food.

4. Please explain why the dispersal assay is for 1 hr, while the exploration defect is for 20 hr.

5. The data shown in graphs should also be in tables in supplementary information.

Reviewer #3: In their manuscript the authors investigate the role of G Protein-Coupled Receptor Kinase-2 (GRK-2) in exploration and quiescence behavior. They use behavioral and other assays in C. elegans in combination with genetics and rescue experiments as well as microscopy to delineate a model for how grk-2 might be working in this system to control exploration behavior and quiescence. The authors found that GRK-2 acts in olfactory sensory neurons to promote exploration and suppress movement quiescence. From the data they conclude that GRK-2 regulates

exploration behavior in an opposite manner to NPR-1 and the neuropeptides FLP-1 and FLP-18. They study the secretion of FLP-1, a neuropeptide that regulates locomotion and found that it depends on GRK-2. Overexpression of flp-1 in turn was found to inhibit exploration behavior.

The manuscript is well written and experiments are rather well documented and the overall story makes sense. The manuscript contains a number of interesting observations and well-conducted experiments and should be interesting for many researchers interested in G protein signaling and behavior. I support publication and have very few comments regarding the manuscript, most of which are rather minor.

1

The authors found that the ocr-2 canonical null mutant ak47 showed an exploration defect that was not observed with the ocr-2(ok1711) allele which the authors presume to also be a null allele. The authors assume that one of the strains might contain a background mutation. ok1711 is not listed in the strain list in the supplement so it is not clear which strain was used and whether this was backcrossed. Did the authors try to backcross the alleles, in an attempt to remove the background mutation? Alternatively, is there a third allele that could be used to clarify the situation? The current situation of leaving this question unresolved appears to be very dissatisfying. Also, please double check that all strains are listed in the supplement.

2

GRK-2 and EGL-4 are proposed to act in opposite ways in the ciliated neurons to regulate exploration. Could the authors explain or speculate how this might happen mechanistically?

3

The authors state that aptf-1 mutants partially suppressed both the enhanced quiescence and reduced activity phenotypes of grk-2 mutants (Fig 5B-D). They concluded that increased sleep is partially responsible for the reduced exploration phenotype of grk-2 mutants.

Based on their data they suggested that GRK-2 acts independently of FLP-11 in regulating quiescence, and that the role of APTF-1 in adult movement quiescence is partially independent of FLP-11.

“Surprisingly, we found that flp-11 mutants do not suppress the enhanced quiescence of grk-2 mutants, suggesting that APTF-1 plays a role in adult movement quiescence independent of its role in promoting FLP-11 expression. One possible explanation is that some sleep-promoting functions of RIS (or other neurons affected by aptf-1 function) do not depend on flp-11.?

The authors sould consider an alternative interpretation of their data: the effect of flp-11(-) is known to be weaker than that of aptf-1(-) (e.g. Turek et al eLife). Hence an alternative explanation could be, that the assay is sensitive enough to detect a phenotype for aptf-1(-) but not for flp-11(-). This potential explanation could be especially relevant since the effect of aptf-1(-) in the assay is already rather modest. In s4c there is a trend of reduced quiescence in flp-11(-), while not scored as statistically significant, this observation supports a role for flp-11 in this process. It is usually recommended to be careful with interpreting the absence of evidence as evidence for absence.

4

“Fig 5. Mutations in aptf-1 partially suppress the exploration defect, movement

quiescence, and reduced activity of grk-2 mutants.

Fig 6. Mutations in npr-1 suppress the exploration defect of grk-2 mutants.

Fig 7. Mutations in flp-1 partially suppress the exploration defect of grk-2 mutants.”

Repeatedly the authors write about “Mutations” when only one allele is tested. In several cases I could not find additional alleles. If only one allele is used for the experiment it would be more precise to state “mutation” as a singular.

5

The authors might want to change their nomenclature to the more common practice of adding the p for promoter behind the gene name, ie Prab-3 to rab-3p? See also:

https://wormbase.org//about/userguide/nomenclature

**Have all data underlying the figures and results presented in the manuscript been provided?**

Reviewer #1: **No: **there are no spreadsheets that contain the numerical data presented in the figures

Reviewer #2: **No: **Data in graphs should also be in tables.

Reviewer #3: Yes

PLOS authors have the option to publish the peer review history of their article (what does this mean?). If published, this will include your full peer review and any attached files.

Reviewer #1: No

Reviewer #2: No

Reviewer #3: No

---

## [Decision Letter · Decision Letter 1]

15 Dec 2022

Dear Dr. Topalidou,

Thank you very much for submitting your Research Article entitled 'G Protein-Coupled Receptor Kinase-2 (GRK-2) regulates exploration through neuropeptide signaling in Caenorhabditis elegans' to PLOS Genetics.

The manuscript was fully evaluated at the editorial level and by independent peer reviewers. The reviewers appreciated the attention to an important topic but identified some concerns that we ask you address in a revised manuscript.

We therefore ask you to modify the manuscript according to the review recommendations made by Reviewer #2.  In particular, the concern of the relatively small "n" of many of the experiments was raised as a concern.  If raising the n to closer to 30, as suggested, is not feasible in all instances, please provide a robust justification for lower sample sizes.  It would also be helpful to note how many transgenic lines were assayed and pooled together in each case, as pooling data from multiple lines is generally considered more rigorous than following the behavior of just a single line that may not be fully representative of the phenotype (due to varied expression levels and mosaicism among arrays/lines).

Please also work to incorporate the suggested changes to improve the clarity of Figure 9.  If some of the suggested changes are not feasible to incorporate into the figure itself, the points (e.g. speculation on the role of the additional neurons) could be added to the text of the manuscript or the figure legend.  The remaining points raised by the reviewer should be able to be incorporated into the text relatively easily, and should be addressed to the extent possible.

Yours sincerely,

Denise M. Ferkey

Guest Editor

PLOS Genetics

Gregory P. Copenhaver

Editor-in-Chief

PLOS Genetics

Reviewer's Responses to Questions

**Comments to the Authors:**

Reviewer #1: The authors have done a good job at revising the manuscript. They have responded very well to all comments, resulting in an improved manuscript. I recommend publication of the manuscript in PLoS Genetics.

Reviewer #2: The authors examine the role of GRK-2 kinase in exploration and quiescence in C. elegans. This is a revised manuscript that addressed many of the previous concerns from the reviewers. However, in including information about how many trials were conducted and how many animals assayed in this revised version, there are several further questions that are raised. The small number of animals assayed and the small number of trials performed is somewhat surprising given that C. elegans is relatively easy to obtain large numbers and some of the assays are not onerous to perform. Generally, at least 30 animals should be assayed to accurately reflect a population, not the 5-40 shown for many strains. In the data presented in Supplementary Table 3, I assume that data were entered for the different trials sequentially (e.g., the first 10 are from trial 1 and the second 10 are from trial 2), so that the data from different trials are not bimodal?

1. Fig. 2C: Why is there such variability in the off food response? This proposed explanation should be included somewhere.

2. Fig. 2D: grk-2; osm-3;:grk-2 animals are responding as though off food, with large variability among animals. No such response is observed with grk-2 mutants alone. Why? Is the variability due to using transgenic animals? However, these large variabilities were not seen when the same animals were assayed on food rescues.

3. line 157: The authors should define exploration/exploring more clearly upfront. In this section the authors present data that the animals are not moving very much from the point of origin of the assay. This lack of movement could be due to decreased roaming, increased dwelling, increased quiescence, etc. The authors lump all of these process into exploration. Is this intentional? The title for this section assumes that defective exploration, which the authors define in line 251 (i.e., much later) is anything that changes movement, and then later includes quiescence into exploration (line 468). However, while their data are suggestive, the data are not sufficient yet to lump all these behaviors together.

4. If the same data for wild type and grk-2 mutants are shown for different figures, it should be indicated in the figures, not just in the Supplemental Tables.

5. line 438: The implication is that the authors are proposing that grk-2 and egl-4 are acting in the same chemosensory neurons; this implication should be explicitly stated as in later statements, they propose an alternative hypothesis whereby the two kinases act in different neurons.

6. Fig. 9 is still confusing. Make the cells larger and show what genes/proteins are acting in the respective cells. For instance, GRK-2 phosphorylates DOP-3, a receptor (which should be illustrated as a transmembrane protein), in premotor interneurons to affect crawling and swimming. The authors are proposing that GRK-2, a kinase, antagonizes the action of another kinase, EGL-4, in ciliated neurons; show that in the cytoplasm as opposed to outside the cell. Among the ciliated neurons that rescue the grk-2 phenotypes, only AWB and ASH have gap junctions with RMG and only ASH also has chemical synapses with RMG. Are the authors proposing that these are the chemosensory neurons in which grk-2 is acting? That is not even indicated in the figure. How is AVK affecting the activity of motor neurons? How does grk-2 activity regulate NPR-1 activity if they are acting in different cells? Is RMG considered in the flp-1 circuit? Fig. 9 remains confusing and too simplistic.

Overall, the authors present interesting results. The numbers of animals, numbers of trials, and the statistics should be strengthened before publication.

Reviewer #3: The authors have addressed all my comments and I support publication.

**Have all data underlying the figures and results presented in the manuscript been provided?**

Reviewer #1: Yes

Reviewer #2: Yes

Reviewer #3: Yes

PLOS authors have the option to publish the peer review history of their article (what does this mean?). If published, this will include your full peer review and any attached files.

Reviewer #1: No

Reviewer #2: No

Reviewer #3: **Yes: **Henrik Bringmann

---

## [Editor Report · Decision Letter 2]

12 Jan 2023

Dear Dr. Topalidou,

We are pleased to inform you that your manuscript entitled "G Protein-Coupled Receptor Kinase-2 (GRK-2) controls exploration through neuropeptide signaling in Caenorhabditis elegans" has been editorially accepted for publication in PLOS Genetics. Congratulations!

Yours sincerely,

Denise M. Ferkey

Guest Editor

PLOS Genetics

Gregory P. Copenhaver

Editor-in-Chief

PLOS Genetics

Comments from the reviewers (if applicable):

**Data Deposition**

http://datadryad.org/submit?journalID=pgenetics&manu=PGENETICS-D-22-00899R2

**Press Queries**

---

## [Editor Report · Acceptance letter]

15 Jan 2023

PGENETICS-D-22-00899R2 

G Protein-Coupled Receptor Kinase-2 (GRK-2) controls exploration through neuropeptide signaling in *Caenorhabditis elegans*

Dear Dr Topalidou, 

We are pleased to inform you that your manuscript entitled "G Protein-Coupled Receptor Kinase-2 (GRK-2) controls exploration through neuropeptide signaling in *Caenorhabditis elegans*" has been formally accepted for publication in PLOS Genetics! Your manuscript is now with our production department and you will be notified of the publication date in due course.

With kind regards,

Zsofia Freund

PLOS Genetics

On behalf of:
